# Computational Efficiency under Covariate Shift in Kernel Ridge Regression

**Andrea Della Vecchia**
Swiss Finance Institute (SFI)
EPFL
andrea.dellavecchia@epfl.ch

**Arnaud Mavakala Watusadisi**
MaLGa, DIBRIS
University of Genova
5775617@studenti.unige.it

**Ernesto De Vito**
MaLGa, DIMA
University of Genova
ernesto.devito@unige.it

**Lorenzo Rosasco**
MaLGa, DIBRIS
University of Genova
Istituto Italiano di Tecnologia
lorenzo.rosasco@unige.it

## Abstract

This paper addresses the covariate shift problem in the context of nonparametric regression within reproducing kernel Hilbert spaces (RKHSs). Covariate shift arises in supervised learning when the input distributions of the training and test data differ, presenting additional challenges for learning. Although kernel methods have optimal statistical properties, their high computational demands in terms of time and, particularly, memory, limit their scalability to large datasets. To address this limitation, the main focus of this paper is to explore the trade-off between computational efficiency and statistical accuracy under covariate shift. We investigate the use of random projections where the hypothesis space consists of a random subspace within a given RKHS. Our results show that, even in the presence of covariate shift, significant computational savings can be achieved without compromising learning performance.

## 1  Introduction

Classical supervised learning assumes that the training and test data distributions are identical Vapnik (1999). However, in practice, it is often the case that there is a significant mismatch between the two Quiñonero-Candela et al. (2022). This discrepancy can arise from various factors, such as inconsistencies in measurement equipment, differences in data collection domains, or variations in subject populations Koh et al. (2021). Among these scenarios, one particularly interesting and common case is known as covariate shift Sugiyama & Kawanabe (2012). Covariate shift occurs when the marginal distributions of the input covariates differ between the training and test data, while the conditional distribution of the output label given the input covariates remains unchanged Shimodaira (2000); Cortes et al. (2008). This phenomenon is observed in a variety of well-studied learning problems, including among all domain adaptation Ben-David et al. (2006); Mansour et al. (2009); Cortes & Mohri (2014); Zhang et al. (2012), active learning Wiens (2000); Kanamori & Shimodaira (2003); Sugiyama & Ridgeway (2006), natural language processing Jiang & Zhai (2007), and medical image analysis Guan & Liu (2021).

Despite its practical significance, covariate shift remains relatively underexplored in theoretical frameworks compared to the classical setting where no distribution mismatch is assumed. Recently, several studies have attempted to bridge this gap. A widely adopted approach to addressing covariate shift involves correcting the learning objective by reweighting the loss function using the so-called

39th Conference on Neural Information Processing Systems (NeurIPS 2025).

importance weighting (IW) function. This function corresponds to the Radon-Nikodym derivative of the test marginal distribution with respect to the training marginal distribution Huang et al. (2006); Sugiyama et al. (2012); Fang et al. (2020). Shimodaira (2000) was the first to demonstrate the consistency of the importance-weighted maximum likelihood estimator, while in Cortes et al. (2010) the authors derived suboptimal finite-sample bounds for restricted function classes with finite pseudodimension. In the context of nonparametric regression in reproducing kernel Hilbert spaces (RKHSs) and under the assumption that the regression function belongs to the RKHS (well-specified case), Gizewski et al. (2022) provided optimal excess risk convergence results for the minimizer of the reweighted empirical risk when the weight function is known and uniformly bounded. In Ma et al. (2023), authors recently showed that the standard unweighted kernel ridge regression estimator is still minimax optimal under an appropriate choice of the regularization parameter, provided the IW function is uniformly bounded or its second moment is bounded. Similar results for nonparametric classification were obtained in Kpotufe & Martinet (2021). We also mention Schmidt-Hieber & Zamolodtchikov (2024); Pathak et al. (2022) in the context of nonparametric regression for classes of functions different from RKHSs, and Wen et al. (2014); Lei et al. (2021); Yamazaki et al. (2007) for parametric models. Building on Ma et al. (2023), in Gogolashvili et al. (2023) the authors extended the analysis to (simplified) misspecified case, where the regression function is not assumed to lie in the RKHS itself, but its projection is. In this setting, the authors showed that, under covariate shift, the unweighted classic KRR predictor is not a consistent estimator of the projection of the regression function. In such cases, IW correction is necessary.

Kernel methods provide a robust framework for nonparametric learning, but their scalability is limited by high computational and memory costs—challenges that clearly do not disappear under covariate shift. To address this, researchers have developed more efficient strategies, ranging from improved optimization (Johnson & Zhang, 2013; Schmidt et al., 2017) to randomized linear algebra techniques (Mahoney, 2011; Drineas & Mahoney, 2005; Woodruff, 2014; Calandriello et al., 2017). These aim to reduce costs, raising a key question: do such shortcuts compromise statistical accuracy? Recent work suggests they often do not (Rudi et al., 2015; Bach, 2017; Bottou & Bousquet, 2008; Sun et al., 2018; Rudi & Rosasco, 2017; Della Vecchia et al., 2021). A promising approach is to restrict the hypothesis space to a lower-dimensional (random) subspace, as in sketching (Kpotufe & Sriperumbudur, 2019) and random projection methods (Woodruff, 2014), including Nyström methods for kernels (Smola & Schökopf, 2000; Williams & Seeger, 2001). Recent results confirm that these methods can achieve both computational efficiency and statistical accuracy, for both smooth and general convex losses (Rudi et al., 2015; Bach, 2013; Marteau-Ferey et al., 2019; Della Vecchia et al., 2024).

Although random projection techniques—such as the Nyström method—have been widely studied in standard learning settings, their use under covariate shift remains largely unexamined. A recent step in this direction was taken by Myleiko & Solodky (2024), building on the framework of Gizewski et al. (2022). The authors consider Nyström subsampling in a setting where weights are known and uniformly bounded, and derive optimal risk bounds that, in the case of Tikhonov regularization, can be recovered as a special case of our results.

**Contribution.** In this paper, we investigate the importance-weighting correction for kernel ridge regression under the covariate shift setting. Unlike most of the previous work in this setting, we employ random projection techniques, specifically the Nyström method, to improve the algorithm's scalability and efficiency. Our primary goal is to explore the balance between computational efficiency and statistical accuracy in this framework. By analyzing the interplay among regularization, subspace size, and the various parameters that characterize the complexity of the problem, we identify the conditions under which the best-known statistical guarantees can be achieved while drastically reducing computational costs. Notably, we show that our algorithm achieves a remarkable result: despite Nyström approximation, it attains the best rates reported in the literature without compromising statistical accuracy, all while being significantly more computationally efficient. Regarding the theoretical aspects, while we build upon established results from the random projections literature, our novel proofs presented here address additional technical challenges arising from the mismatch between training and test distributions, as well as the potential unboundedness of the weighting function in the empirical risk minimization. Full technical details are provided in Appendix B. Finally, we validate these theoretical findings through simulations and real data experiments.

**Limitations.** As remarked in Section 5, Assumption 3 imposes boundedness of all moments of the weight function. While this is a relatively strong condition, especially when compared to the mere second-moment boundedness required in Ma et al. (2023), it is essential in our setting to ensure

the applicability of the Nyström method with ALS sampling in the unbounded case. Assumption 2 ensures that the projection of the regression function onto the hypothesis space $\mathcal{H}$ exists and lies within $\mathcal{H}$, rather than on its boundary. This is slightly more general than the standard well-specified setting, but conceptually similar. We do not consider the fully misspecified case here, which introduces significant additional complexity (see interpolation spaces in Steinwart & Christmann (2008) for example). Finally, as mentioned in Section 8, a more refined understanding of the role of the constants in the learning rate bounds could explain why, in practice, a misalignment between training and test distributions can sometimes be benign or, conversely, highly adversarial, depending on its relation with the source condition. These directions are left for future work.

**Organization.** The paper is organized as follows. In Section 2, we introduce the key definitions and notations used throughout the paper, while refreshing KRR in the classical setting. Section 3 provides an overview of KRR in the covariate shift setting. In Section 4, we present empirical risk minimization on random subspaces using Nyström. Section 5 explores the interplay between computational efficiency and statistical accuracy. Section 6 extends our analysis to scenarios where the true IW function is unknown. Section 7 presents a series of simulations and real-world experiments.

## 2 Background

We start defining some key quantities we will need in the rest of the paper, see e.g. Caponnetto & De Vito (2007); Smale & Zhou (2007). Given a measurable space $\mathcal{X}$, a probability distribution $\mu$ on $\mathcal{X}$, a space of square-integrable functions $L_\mu^2$ with respect to measure $\mu$ and a Reproducing Kernel Hilbert Space (RKHS) $\mathcal{H}$ of (bounded) kernel $K : \mathcal{X} \times \mathcal{X} \to \mathbb{R}$, with $K_x(\cdot) = K(x, \cdot) \in \mathcal{H}$, define

$$S_\mu : \mathcal{H} \to L_\mu^2, \quad (S_\mu f)(x) = \langle f, K_x \rangle_\mathcal{H} = f(x) \quad \mu - a.s. \quad \text{and} \quad S_\mu^* g = \int K_x g(x) d\mu.$$

for $f \in \mathcal{H}$, $g \in L_\mu^2$. Covariance operator $\Sigma_\mu : \mathcal{H} \to \mathcal{H}$ is defined as $\Sigma_\mu := S_\mu^* S_\mu = \mathbb{E}_\mu[K_x \otimes K_x]$. Define the sampling operator $\widehat{S} : \mathcal{H} \to \mathbb{R}^n$ associated with set $\{x_1, \ldots, x_n\} \in \mathcal{X}^n$, for $f \in \mathcal{H}$, as

$$(\widehat{S} f)_i := f(x_i) = \langle f, K_{x_i} \rangle_\mathcal{H}, \ \ i \in [n], \quad \text{and} \quad \widehat{S}^*(\widehat{y}) := \frac{1}{n} \sum_{i=1}^n y_i K_{x_i}, \quad \widehat{y} \in \mathbb{R}^n.$$

### 2.1 Classical Setting

Classical nonparametric regression aims to predict a real-valued output $\mathcal{Y} = \mathbb{R}$ given a vector of covariates $X \in \mathcal{X}$. More formally, let $\mathcal{X} \times \mathbb{R}$ be a probability space with distribution $\rho$, where $\mathcal{X}$ and $\mathbb{R}$ are the input and output spaces, respectively. Let $\rho_X$ denote the marginal distribution of $\rho$ on $\mathcal{X}$ and $\rho(\cdot|x)$ the conditional distribution on $\mathbb{R}$ given $x \in \mathcal{X}$. For any fixed $x \in \mathcal{X}$, the optimal estimator in a mean-squared sense is given by the regression function $g^*(x) := \int y d\rho(y|x)$, i.e.

$$g^* = \arg\min_{g \in \mathbb{R}^\mathcal{X}} \mathcal{R}(g) = \mathbb{E}[(Y - g(X))^2].$$

The minimization problem is generally unsolvable since the distribution $\rho$ is unknown. In practice, we only have access to a dataset of $n$ input-output pairs $(x_i, y_i)_{i=1}^n \in (\mathcal{X} \times \mathbb{R})$ sampled independently and identically (i.i.d.) from the joint distribution $\rho$, i.e. $(x_1, y_1), \ldots, (x_n, y_n) \sim \rho(x, y)$.

A common approach to derive an approximation of $g^*$ involves restricting the hypothesis space from all measurable functions to a real separable Hilbert space $\mathcal{H}$, and replacing the expected risk $\mathcal{R}$ with the (regularized) empirical risk $\widehat{\mathcal{R}}_\lambda : \mathcal{H} \to [0, \infty)$ defined as

$$\widehat{\mathcal{R}}_\lambda(f) := \frac{1}{n} \sum_{i=1}^n (y_i - f(x_i))^2 + \lambda \|f\|_\mathcal{H}^2, \quad f \in \mathcal{H}, \quad \lambda > 0. \tag{1}$$

The (Regularized) Empirical Risk Minimization (ERM) algorithm then solves:

$$\widehat{f}_\lambda := \arg\min_{f \in \mathcal{H}} \widehat{\mathcal{R}}_\lambda. \tag{2}$$

When $\mathcal{H}$ is an RKHS, as in the rest of this paper, the resulting estimator is known as the kernel ridge regression (KRR) estimator. Our primary goal is to upper bound the excess risk $\mathcal{E}$ of $\widehat{f}_\lambda$, i.e.

$$\mathcal{E}(\widehat{f}_\lambda) := \mathcal{R}(\widehat{f}_\lambda) - \mathcal{R}(g^*) = \|\widehat{f}_\lambda - g^*\|_{\rho_X}^2.$$

Here, the equality follows from a standard result, see for example Caponnetto & De Vito (2007).

# 3 Covariate Shift Setting

The covariate shift setting introduces an additional complexity: training and test distributions may differ, but only through their marginals, while sharing the same conditional distribution:

$$\rho^{te}(x, y) = \rho(y|x)\rho_X^{te}(x), \qquad \rho^{tr}(x, y) = \rho(y|x)\rho_X^{tr}(x).$$

Since the regression function $g^*$ only depends on the conditional distribution, which is identical for both $\rho^{te}$ and $\rho^{tr}$, it is unique. As in the standard setting, we are provided with $n$ input-output pairs $(x_i, y_i)_{i=1}^n \in (\mathcal{X} \times \mathbb{R})$ sampled i.i.d. from $\rho^{tr}$, i.e. $(x_1, y_1), \ldots, (x_n, y_n) \sim \rho^{tr}(x, y)$.

The challenge consists in the fact that we train our model using samples from the $\rho^{tr}$, but we aim to evaluate its performance on new data drawn from $\rho^{te}$. Then, we want to upper bound the excess risk

$$\mathcal{E}(\widehat{f}_\lambda) = \|\widehat{f}_\lambda - g^*\|_{\rho_X^{te}}^2.$$

Note that the empirical risk computed from $\rho^{tr}$ samples is a biased estimate of the expected risk under $\rho^{te}$. As a result, minimizing it may not yield a predictor that performs well on the test distribution.

## 3.1 Importance-Weighting (IW) Correction

The goal of importance-weighting (IW) correction is to construct an unbiased estimator of the risk with respect to test distribution $\rho^{te}$, while using data sampled from $\rho^{tr}$. The idea is to reweight ERM samples based on their *relevance* to the test distribution, ensuring good performance under $\rho^{te}$. If $\rho_X^{te} \ll \rho_X^{tr}$, we can define the IW function $w$ representing the weight assigned to a point $x \in \mathcal{X}$ as the Radon-Nikodym derivative of $\rho_X^{te}$ with respect to $\rho_X^{tr}$:

$$w(x) := \frac{d\rho_X^{te}}{d\rho_X^{tr}}(x). \tag{3}$$

Points that are likely to be encountered during testing ($\rho_X^{te}$ is large) while are rare to be sampled at training time ($\rho_X^{tr}$ is small) are considered particularly relevant and will receive higher weights.

In the rest of the paper, we will focus on applying this framework in the context of kernel methods.

**Assumption 1.** *$\mathcal{H}$ is an RKHS with scalar product $\langle \cdot, \cdot \rangle_\mathcal{H}$ and associated kernel $K : \mathcal{X} \times \mathcal{X} \to \mathbb{R}$.*

We define the regularized importance-weighted empirical risk, for all $f \in \mathcal{H}$, as

$$\widehat{\mathcal{R}}_\lambda^w(f) := \frac{1}{n} \sum_{i=1}^n w(x_i)(y_i - \langle K_{x_i}, f \rangle)^2 + \lambda\|f\|_\mathcal{H}^2, \tag{4}$$

where $\mathcal{H} \ni K_x(\cdot) = K(x, \cdot)$. For the square loss, the minimizer $\widehat{f}_\lambda^w$ of eq. (4) is given by:

$$\widehat{f}_\lambda^w(x) = (\widehat{S}^*\widehat{M}_w\widehat{S} + \lambda\mathrm{I})^{-1}\widehat{S}^*\widehat{M}_w\widehat{y}, \tag{5}$$

where $\widehat{M}_w$ is the diagonal matrix with $i$-th entry $w(x_i)$. In case weights are all positive, we have

$$\widehat{f}_\lambda^w(x) = \sum_{i=1}^n c_i^w K_{x_i}(x) \in \mathrm{span}\{K_{x_1}, \ldots, K_{x_n}\}, \quad c^w = (\widehat{K} + n\lambda\widehat{M}_{1/w})^{-1}\widehat{y} \in \mathbb{R}^n, \tag{6}$$

where $\widehat{K}$ is the kernel Gram matrix, and $\widehat{M}_{1/w}$ is the diagonal matrix with $i$-th entry $1/w(x_i)$. Since the regression function $g^*$ may not generally belong to $\mathcal{H}$ (i.e., the model may be misspecified), we introduce the best approximation $f_\mathcal{H} \in \mathcal{H}$ of $g^*$ with respect to the $L_{\rho_X^{te}}^2$ distance.

**Assumption 2.** *There exists an $f_\mathcal{H} \in \mathcal{H}$ such that*

$$\mathcal{E}(f_\mathcal{H}) = \min_{f \in \mathcal{H}} \mathcal{E}(f) = \min_{f \in \mathcal{H}} \|f - g^*\|_{\rho_X^{te}}^2. \tag{7}$$

Note that, while the minimizer might not be unique, we select $f_\mathcal{H}$ as the unique minimizer with minimal norm (De Vito et al., 2021). In the following, we will evaluate the performance of our estimator relative to the best estimator in $\mathcal{H}$, i.e. $f_\mathcal{H}$.

**Computations**  A significant limitation of the procedure outlined above is the computational cost associated with the $n \times n$ matrix inversion required to compute the estimator, see (6). This operation has a complexity of $\mathcal{O}(n^3)$ in time and $\mathcal{O}(n^2)$ in memory, making it impractical when $n > 10^5$.

# 4  ERM on Random Subspaces and the Nyström Method

In this paper, we consider an efficient approximation of the above procedure based on considering a subspace $\mathcal{B} \subset \mathcal{H}$ and solving the corresponding importance-weighted regularized ERM problem

$$\min_{\beta \in \mathcal{B}} \widehat{\mathcal{R}}_\lambda^w(\beta), \tag{8}$$

with $\widehat{\beta}_\lambda^w$ the unique minimizer. As clear from (6), choosing $\mathcal{B} = \mathcal{H}_n = \operatorname{span}\{K_{x_1}, \ldots, K_{x_n}\}$ is equivalent to considering the full space $\mathcal{H}$ and yields the same solution as in (5). However, a natural alternative is to consider a smaller subspace:

$$\mathcal{B} = \mathcal{H}_m = \operatorname{span}\{K_{\widetilde{x}_1}, \ldots, K_{\widetilde{x}_m}\}, \tag{9}$$

where $\{\widetilde{x}_1, \ldots, \widetilde{x}_m\} \subset \{x_1, \ldots, x_n\}$ is a random subset of the input points and $m \leq n$. This is equivalent to Nyström approximation (Williams & Seeger, 2000). A basic approach is to select these points uniformly at random from the training dataset. Alternatively, we can employ more refined sampling techniques, such as using leverage scores (Drineas et al., 2012)

$$l_i(\alpha) = (\widehat{K}(\widehat{K} + \alpha n \mathrm{I})^{-1})_{ii}, \qquad i = 1, \ldots, n. \tag{10}$$

Since in practice computing leverage scores directly can be computationally expensive, approximations $(\hat{l}_i(\alpha))_{i=1}^n$ have been considered (Drineas et al., 2012; Cohen et al., 2015; Alaoui & Mahoney, 2015). In particular, we consider the following one.

**Definition 1** (*T-approximate leverage scores*)**.** *Let $(l_i(\alpha))_{i=1}^n$ be the leverage scores associated to the training set for a given $\alpha$. Let $\delta > 0$, $t_0 > 0$ and $T \geqslant 1$. We say that $(\hat{l}_i(\alpha))_{i=1}^n$ are $T$-approximate leverage scores with confidence $\delta$, when with probability at least $1 - \delta$,*

$$\frac{1}{T} l_i(\alpha) \leqslant \hat{l}_i(\alpha) \leqslant T l_i(\alpha), \qquad \forall i \in \{1, \ldots, n\}, \; \alpha \geqslant t_0. \tag{11}$$

Given the $T$-approximate leverage scores for $\alpha \geqslant t_0$, ALS sampling proceeds by independently drawing samples $\tilde{x}_1, \ldots, \tilde{x}_m$ from the training set with replacement, where each point $x_i$ is selected with probability $Q_\alpha(i) = \hat{l}_i(\alpha)/\sum_j \hat{l}_j(\alpha)$. All the results in the next sections are obtained under ALS sampling.

We can now define the Nyström W-KRR problem as follows:

$$\widehat{f}_{\lambda,m}^w := \operatorname*{arg\,min}_{\beta \in \mathcal{H}_m} \frac{1}{n} \left\| \widehat{M}_w^{1/2}(\widehat{y} - \widehat{S}\beta) \right\|_2^2 + \lambda \|\beta\|_{\mathcal{H}}^2 = \operatorname*{arg\,min}_{f \in \mathcal{H}} \frac{1}{n} \left\| \widehat{M}_w^{1/2}(\widehat{y} - \widehat{S}P_m f) \right\|_2^2 + \lambda \|f\|_{\mathcal{H}}^2, \tag{12}$$

where $P_m$ is the orthogonal projection operator onto $\mathcal{H}_m$, given by $P_m = VV^*$, see Appendix A. Taking the derivative and using the first order condition (see Appendix A), the Nyström estimator can be expressed as:

$$\widehat{f}_{\lambda,m}^w = V(V^*\widehat{S}^*\widehat{M}_w\widehat{S}V + \lambda \mathrm{I})^{-1}V^*\widehat{S}^*\widehat{M}_w\widehat{y}. \tag{13}$$

Alternatively, using some linear algebra, it can also be written as:

$$\widehat{f}_\lambda^w(x) = \sum_{i=1}^m \widetilde{c}_i^w K_{\widetilde{x}_i}(x), \qquad \widetilde{c}^w = (\widehat{K}_{nm}^T \widehat{M}_w \widehat{K}_{nm} + n\lambda \widehat{K}_{mm})^{-1} \widehat{K}_{nm}^T \widehat{M}_w \widehat{y}, \tag{14}$$

where $\widehat{f}_\lambda^w(x) \in \operatorname{span}\{K_{\widetilde{x}_1}, \ldots, K_{\widetilde{x}_m}\}$, $\widetilde{c}^w \in \mathbb{R}^m$, $\widehat{K}_{nm} \in \mathbb{R}^{n \times m}$, $(\widehat{K}_{nm})_{ij} = K(x_i, \widetilde{x}_j)$ and $\widehat{K}_{mm} \in \mathbb{R}^{m \times m}$, $(\widehat{K}_{mm})_{ij} = K(\widetilde{x}_i, \widetilde{x}_j)$ (see derivation in Appendix A).

**Computations**  From eq. (14), it is clear that the Nyström method can offer significant computational benefits. Unlike in eq. (13), computing our projected estimator only requires $\mathcal{O}(m^3 + m^2 n)$ time and $\mathcal{O}(mn)$ memory, compared to the previous $\mathcal{O}(n^3)$ and $\mathcal{O}(n^2)$. When $m \ll n$ the difference between the two can be big and efficient implementations such as in Rudi et al. (2017); Meanti et al. (2020) can drastically reduce computational requirements.

# 5 Statistical Guarantees

In this section, we aim to derive excess risk bounds for the Nyström estimator presented in eq. (13). We begin by introducing the technical assumptions required for the subsequent analysis.

## 5.1 Further Assumptions

The following assumption, inspired by Gogolashvili et al. (2023), ensures the boundedness of the importance-weighting (IW) function or of its moments.

**Assumption 3.** *Let $w = d\rho_X^{te}/d\rho_X^{tr}$ be the IW function. There exist constants $q \in [0,1], W > 0$ and $\sigma > 0$ such that $\forall p \in \mathbb{N}, \ p \geqslant 2$*

$$\left( \int_{\mathcal{X}} w(x)^{\frac{p-1}{q}} d\rho_X^{te} \right)^q \leqslant \frac{1}{2} p! W^{p-2} \sigma^2, \tag{15}$$

*where the left-hand side for $q = 0$ is defined as $\left\| w^{p-1} \right\|_{\infty, \rho_X^{te}}$, the* ess sup *with respect to $\rho_X^{te}$.*

Considering the uniformly bounded case $\|w\|_\infty < \infty$, Assumption 3 is satisfied for $q = 0$. When $w$ is not uniformly bounded, Assumption 3 can still be satisfied for $q \in (0,1]$ if the moments of $w$ are bounded. For example, it is satisfied for $q \in (0,1]$ if $W \geqslant 1, \sigma^2 \geqslant 1$ and

$$2\rho_X^{te} \left( \left\{ x \in X : \frac{d\rho_X^{te}}{d\rho_X^{tr}}(x) \geqslant t \right\} \right) \leqslant \sigma^2 \exp\left( -W^{-1} t^{1/q} \right) \quad \text{ for all } t > 0$$

(see Appendix A in Gogolashvili et al. (2023) for the detailed result). Equivalently, Assumption 3 can be stated as a condition on the Rényi divergence between $\rho_X^{te}$ and $\rho_X^{tr}$ (Mansour et al., 2009; Cortes et al., 2010). It follows that this assumption can be interpreted as a requirement for the test distribution $\rho_X^{te}$ not to deviate significantly from the train distribution $\rho_X^{tr}$, with $q \in [0,1]$ quantifying the extent of the deviation. Introducing the parameter $q$ is useful for presenting results such as Theorem 1 in a unified form, which can then be specialized to the two cases considered in Corollary 1: uniformly bounded ($q = 0$) and possibly unbounded ($q \neq 0$) weights. Note that in Ma et al. (2023), a weaker assumption requiring only the second moment to be bounded is considered. The stronger condition in eq. (15) is anyway crucial in our case to apply the Nyström method in the unbounded setting (see Appendix B and how the proof differentiates from Rudi et al. (2015)).

**Assumption 4.** *The range of the output $Y \in \mathbb{R}$ is upper bounded, i.e. $Y \in [-B, B], \ B < \infty$.*

Furthermore, we make the following regularity assumption, usually known as *source condition*.

**Assumption 5** (Source condition)**.** *There exist $1/2 \leqslant r \leqslant 1$ and $g \in L^2\left(X, \rho_X^{te}\right)$ with $\|g\|_{\rho^{te}} \leqslant R$ for some $R > 0$ such that $f_{\mathcal{H}} = L^r g$, where $L := S_{\rho^{te}} S_{\rho^{te}}^*$ is the integral operator.*

Assumption 5 and its equivalent formulations (e.g., Assumption 4 in Rudi et al. (2015)) are common in literature (Smale & Zhou, 2007; Caponnetto & De Vito, 2007). $r$ quantifies the smoothness of the target function $f_{\mathcal{H}}$ and the extent to which it can be well approximated by functions in $\mathcal{H}$. For $r = 1/2$, the assumption is always satisfied. Intuitively, a larger $r$ implies $f_{\mathcal{H}}$ being smoother.

Finally, we impose an assumption on the capacity of our RKHS, which roughly measures the number of eigenvalues of $\Sigma$ greater than $\lambda$ (Zhang, 2005; Caponnetto & De Vito, 2007).

**Definition 2** (Effective dimension)**.** *For $\lambda > 0$, define the random variable $\mathcal{N}_x(\lambda) = \left\langle K_x, (\Sigma + \lambda \mathrm{I})^{-1} K_x \right\rangle_{\mathcal{H}}$ with $x \in \mathcal{X}$ distributed according to $\rho$, then $\mathcal{N}_\rho(\lambda) = \mathbb{E}_\rho \mathcal{N}_x(\lambda)$ is called effective dimension.*

**Assumption 6** (Capacity condition)**.** *$\forall \lambda > 0$, there exists $0 \leqslant \gamma \leqslant 1, Q > 0$ s.t. $\mathcal{N}_{\rho_X^{te}}(\lambda) < Q\lambda^{-\gamma}$.*

It is known that the condition in Assumption 6 is ensured if the eigenvalues $(\eta_i)_i$ of the covariance operator $\Sigma$ satisfy a polynomial decaying condition $\eta_i \sim i^{-1/\gamma}$ (see Appendix C).

## 5.2 Excess Risk Bounds

In this section, we present our main theoretical results. We will derive excess risk bounds for the Nyström predictor defined in eq. (13) and we will show that Nyström approximation does not affect the state-of-art rates of convergence while instead reducing both time and memory requirements.

We present now the main result of the paper.

**Theorem 1.** *Under assumptions 1,2,3,4,5,6, for ALS sampling, let $\delta > 0$, $\left(\frac{256(W+\sigma^2)\log^2(4/\delta)}{n}\right)^{\frac{1}{\gamma(1-q)+1+q}} \leqslant \lambda \leqslant \|\Sigma\|_{op}$, and $m \geqslant 144T^2Q\lambda^{-\gamma}\log\frac{8n}{\delta}$, with probability greater or equal than $1 - \delta$*

$$\left|\mathcal{R}(\widehat{f}^w_{\lambda,m}) - \mathcal{R}(f_{\mathcal{H}})\right|^{1/2} \leqslant 64B\left(\frac{W}{n\sqrt{\lambda}} + \sqrt{\frac{\sigma^2}{n\lambda^{\gamma(1-q)+q}}}\right)\log\left(\frac{8}{\delta}\right) + 43R\lambda^r.$$

The detailed proof of Theorem 1 can be found in Appendix B.
If the weighting function $w$ is bounded, i.e. $\|w\|_\infty < \infty$, then Assumption 3 is satisfied for $q = 0$. We specify Theorem 1 for this setting against the unbounded one.

**Corollary 1.** *Under the assumptions and conditions as in Theorem 1,*

*(a) with Assumption 3 satisfied by $q = 0$ and choosing $\lambda \asymp (\|w\|_\infty/n)^{\frac{1}{2r+\gamma}}$, with $m \gtrsim (n/\|w\|_\infty)^{\frac{\gamma}{2r+\gamma}}\log n$, with high probability*

$$\mathcal{E}(f^w_{\lambda,m}) = \|\widehat{f}^w_{\lambda,m} - f_{\mathcal{H}}\|^2_{\rho^{te}_X} \lesssim \left(\frac{\|w\|_\infty}{n}\right)^{\frac{2r}{2r+\gamma}}, \tag{16}$$

*(b) with Assumption 3 satisfied by $q = 1$ and choosing $\lambda \asymp (n/(W+\sigma^2))^{-\frac{1}{2r+1}}$, with $m \gtrsim (n/(W+\sigma^2))^{\frac{\gamma}{2r+1}}\log n$, with high probability*

$$\mathcal{E}(f^w_{\lambda,m}) = \|\widehat{f}^w_{\lambda,m} - f_{\mathcal{H}}\|^2_{\rho^{te}_X} \lesssim \left(\frac{1}{n}\right)^{\frac{2r}{2r+1}}. \tag{17}$$

The rate in eq. (16) matches the optimal convergence rate of standard kernel ridge regression (KRR) established in Caponnetto & De Vito (2007). However, it explicitly depends on $\|w\|_\infty$, which can become arbitrarily large as the training and test distributions diverge. This result also recovers Theorem 2.1 from Myleiko & Solodky (2024) for Tikhonov regularization in the special case $\gamma = 1$. Compared to their work, we consider ALS sampling, which enables fast rates under a suitable capacity condition. Furthermore, we extend the analysis to the case of unbounded importance weights. Specifically, eq. (17) shows that when the weighting function is unbounded (i.e., $q = 1$ in Assumption 3), the convergence rate deteriorates to $\mathcal{O}(n^{-\frac{2r}{2r+1}})$. This slower rate, which does not depend on the capacity assumption 6), is always worse than the rate in eq. (16). These findings are consistent with the results reported in Gogolashvili et al. (2023) for the full (non-projected) model. Note that the rate in eq. (17) can possibly be improved, even under our Nyström approximation, by clipping the unbounded IW function $w$ at a threshold that depends on $n$ (see Section 5.2.2 in Gogolashvili et al. (2023)). This idea follows Corollary 2 of Ma et al. (2023), although the result is not directly comparable due to differing assumptions.

**Example 1.** *To illustrate the benefits of the Nyström approach, consider for example the common setting $q = 0$, $r = 1/2$ and $\gamma = 1$. From eq. (16), we achieve the optimal rate of $\mathcal{O}(n^{-1/2})$ (Caponnetto & De Vito, 2007), with $m = \mathcal{O}(\sqrt{n}\log(n))$. This results in computational costs of $\mathcal{O}(m^3 + m^2n) = \mathcal{O}(n\sqrt{n} + n^2)$ in time and $\mathcal{O}(mn) = \mathcal{O}(n\sqrt{n})$ in memory. These are significantly lower than the costs of the non-approximated method, respectively $\mathcal{O}(n^3)$ and $\mathcal{O}(n^2)$.*

# 6 Unknown Weights

Clearly, when dealing with real data, assuming knowledge of the exact weighting function $w$ is unrealistic. Instead, if we have access to some samples from both distributions, we can attempt to estimate an approximate weighting function $v \approx w$ and control the error resulting from this approximation. Let us choose a function $v$ such that there exists a probability distribution $\rho^v_X$ with

$$v(x) = \frac{d\rho^v_X(x)}{d\rho^{tr}_X(x)}, \tag{18}$$

i.e. $v(x)$ is the Radon-Nikodym derivative of $\rho_X^v$ with respect to $\rho_X^{tr}$. Note that, in general, $\rho_X^v \neq \rho_X^{te}$. Similarly to eq. (7) in Assumption 2, we assume that

$$f_{\mathcal{H}}^v = \arg\min_{f \in \mathcal{H}} \mathcal{E}^v(f) = \|f - g^*\|_{\rho_X^v}^2 \tag{19}$$

exists (again we consider the unique with minimal norm in case of multiple minimizers). Furthermore, we assume that $v$ satisfies Assumptions 3, 5, 6 with constants $V, \eta, r, \gamma, Q$, respectively. Since $v$ is chosen, weights $v(x)$ are known for all $x \in \mathcal{X}$, allowing us to compute the estimator

$$\widehat{f}_{\lambda,m}^v = V(V^*\widehat{S}^*\widehat{M}_v\widehat{S}V + \lambda\mathrm{I})^{-1}V^*\widehat{S}^*\widehat{M}_v\widehat{y}, \tag{20}$$

where $\widehat{M}_v$ is the diagonal matrix with diagonal entries $v(x_1), \dots, v(x_n)$. Still, we are interested in the excess risk of this estimator with respect to the original $\rho^{te}$ distribution.

Denoting $\Sigma = \Sigma_{\rho_X^{te}}$ and $\Sigma_v = \Sigma_{\rho_X^v}$, we decompose the excess risk of $\widehat{f}_{\lambda,m}^v$ as:

$$\mathcal{E}(f_{\lambda,m}^v) = \|\widehat{f}_{\lambda,m}^v - f_{\mathcal{H}}\|_{\rho_X^{te}}^2 \leqslant \|\Sigma\Sigma_{\lambda v}^{-1}\|_{op}\|\widehat{f}_{\lambda,m}^v - f_{\mathcal{H}}^v\|_{\rho_X^v}^2 + \|f_{\mathcal{H}}^v - f_{\mathcal{H}}\|_{\rho_X^{te}}^2. \tag{21}$$

**Well Specified Case**    If $g^* \in \mathcal{H}$, then clearly $g^* = f_{\mathcal{H}} = f_{\mathcal{H}}^v$ and eq. (21) simplifies to:

$$\|\widehat{f}_{\lambda,m}^v - f_{\mathcal{H}}\|_{\rho_X^{te}}^2 \leqslant \|\Sigma\Sigma_{\lambda v}^{-1}\|_{op}\|\widehat{f}_{\lambda,m}^v - f_{\mathcal{H}}\|_{\rho_X^v}^2. \tag{22}$$

The second term corresponds to what we have already analyzed in Theorem 1, with the distribution $\rho_X^{te}$ and its associated weighting function $w$ replaced by $\rho_X^v$ and $v$. The first term quantifies the additional cost incurred due to the mismatch between $\rho_X^{te}$ and $\rho_X^v$. Using Proposition 5 from Appendix C, we can show that if we set $v(x) \equiv 1$, i.e. $\rho_X^v \equiv \rho_X^{tr}$, then $\|\Sigma\Sigma_{\lambda v}^{-1}\|_{op} \leqslant \|w\|_\infty$. This indicates that, when $w$ is bounded, we have a finite control on the appearing term involving the two covariance operators. By setting $\lambda \asymp (\|w\|_\infty n)^{\frac{1}{2r+\gamma}}$, and ensuring $m \gtrsim n^{\frac{\gamma}{2r+\gamma}} \log n$, then, with high probability

$$\|\widehat{f}_{\lambda,m}^{v\equiv 1} - f_{\mathcal{H}}\|_{\rho_X^{te}}^2 \lesssim \|w\|_\infty \left(\frac{1}{n}\right)^{\frac{2r}{2r+\gamma}}, \tag{23}$$

where we used the fact that $\|v\|_\infty = 1$. It is important to note that, when compared with the bound in eq. (16), which assumes knowledge of the true (typically unknown) weights, we achieve the same rate in $n$. This means that, when the model is well-specified, the classical Nyström-ERM algorithm gives the same rate as the importance-weighted variant, despite the covariate shift between train and test distributions. This result agrees with findings in Ma et al. (2023); Gogolashvili et al. (2023), where anyway random projection approximations are not involved. More than that, we emphasize that although the rate remains the same, the dependence on $\|w\|_\infty$, which is assumed finite but can be arbitrarily large, is worse than in eq. (16), where the true $w$ is employed.

**Misspecified Case**    In the misspecified case, the situation is more complex since in general $g^* \neq f_{\mathcal{H}} \neq f_{\mathcal{H}}^v$. In particular, the term $\|f_{\mathcal{H}}^v - f_{\mathcal{H}}\|_{\rho_X^{te}}^2$ in eq. (21) is not zero, and its magnitude can become arbitrarily large, depending on the severity of the mismatch between $\rho_X^{te}$ and $\rho_X^v$.

## 7    Simulations and real data experiments

As emphasized in the introduction, the main goal of this study is to show that the Nyström method can deliver significant computational savings under covariate shift without compromising accuracy.

### 7.1    Simulations

We start by reproducing the experimental setting in Gogolashvili et al. (2023). We want to solve a regression problem using KRR with RBF kernel in the context of distribution shift, assuming $\rho_X^{tr} \sim \mathcal{N}(\mu_{tr}, \Sigma_{tr})$, $\rho_X^{te} \sim \mathcal{N}(\mu_{te}, \Sigma_{te})$ and $\mu_{tr} \neq \mu_{te}, \Sigma_{tr} \neq \Sigma_{te}$. The regression function is

$$g^*(x) := c_1 e^{-\frac{c_2}{\|x\|_2^{2k}}}, \quad k \in \mathbb{N}, \ c \in \mathbb{N},$$

where the parameter $k$ controls the level of misspecification. Note that, in fact, when $k$ increases, the regression function becomes essentially piece-wise constant, and neither constant nor discontinuous

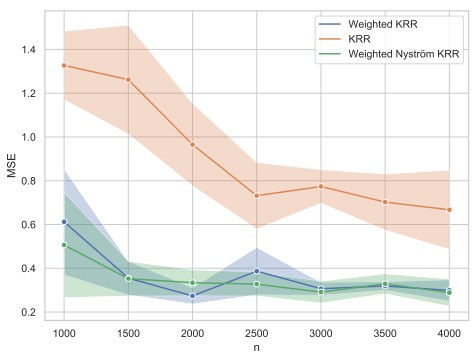 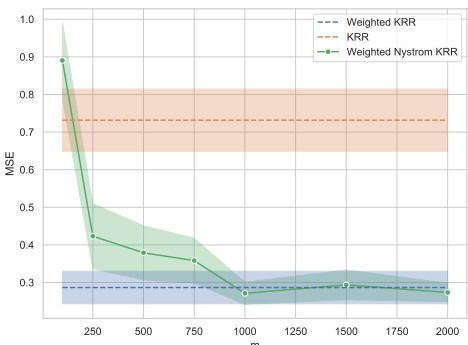

Figure 1: On the left: MSE for the different models varying the number of train samples $n$. The result for Nyström W-KRR model is obtained for optimal $m$. On the right: $n = 3000$, optimal $m$ is selected as the smallest for which Nyström W-KRR matches the full W-KRR model ($m = 1000$ here).

functions belong to the RKHS of the Gaussian kernel. Data samples are generated following $y_i^{tr} = g^*(x_i^{tr}) + \xi_i$, $y_i^{te} = g^*(x_i^{te})$, with $x_i^{tr} \sim \rho_X^{tr}$, $x_i^{te} \sim \rho_X^{te}$ and $\xi_i \sim \mathcal{N}(0, \varepsilon^2)$. Figure 1 shows the results in this setting for $k = 50$. In the plot on the left, we observe that the two weighted models, namely KRR with IW correction (W-KRR) and its Nyström-approximated version (Nyström W-KRR), perform similarly. As expected, the simple (unweighted) KRR model shows a performance gap. On the right, despite reaching the same error of the weighted KRR model, Nyström approximation can lead to important computational savings allowing for choosing a number of Nyström centers $m \ll n$.

## 7.2 Experiments on Benchmark Datasets

As regards real-world applications, we conduct experiments on commonly used benchmark datasets in the domain adaptation field (Wang & Sun, 2024; Wilson et al., 2020; He et al., 2023; Dinu et al., 2023). The size of the original datasets is reduced in case of memory issues with KRR and W-KRR when building the full Gram matrix $\hat{K}$ (see memory bottlenecks in Section 4). In these experiments, weights are estimated using RuLSIF method (Yamada et al., 2013; Liu et al., 2013). We consider 4 real-world datasets: HHAR (Stisen et al., 2015), WISDM (Kwapisz et al., 2011), HAR70+ (Ustad et al., 2023) and HARChildren (Tørring et al., 2024). These datasets consist of data collected from multiple users using wearable sensors, such as accelerometers and gyroscopes. To simulate covariate shift, we train each model on data collected from one user and evaluate it on data from a different user. We employ an RBF kernel with length-scale parameter $\gamma$ and regularization parameter $\lambda$, both selected via cross-validation. The results reported in Table 1 are obtained using ALS sampling (see Definition 1), specifically through the BLESS fast implementation described in Rudi et al. (2018). For comparison, Table 2 in Appendix D presents analogous results obtained with uniform sampling of the Nyström centers. Additional details on the datasets and experimental setup are provided in Appendix D.

Table 1: Performances of the various methods, both in terms of MSE and training/prediction time.

|  | HAR70+ ($n = 20000$) | | | HARChildren ($n = 15000$) | | | HHAR ($n = 15500$) | | | WISDM ($n = 25000$) | | |
|---|---|---|---|---|---|---|---|---|---|---|---|---|
|  | MSE | $t$ train (s) | $t$ pred (s) | MSE | $t$ train (s) | $t$ pred (s) | MSE | $t$ train (s) | $t$ pred (s) | MSE | $t$ train (s) | $t$ pred (s) |
| KRR | $10 \pm 1$ | $1694 \pm 2$ | $15.0 \pm 0.5$ | $26.6 \pm 0.9$ | $762 \pm 12$ | $10.2 \pm 0.4$ | $3.7 \pm 0.3$ | $876 \pm 6$ | $10.5 \pm 0.9$ | $7.8 \pm 0.1$ | $3280 \pm 48$ | $38 \pm 5$ |
| W-KRR | $5.0 \pm 0.2$ | $1785 \pm 2$ | $15.1 \pm 0.3$ | $13.5 \pm 0.8$ | $809 \pm 26$ | $9.0 \pm 0.1$ | $1.8 \pm 0.3$ | $1034 \pm 93$ | $9.9 \pm 0.1$ | $4.8 \pm 0.2$ | $3364 \pm 30$ | $33 \pm 2$ |
| Ny W-KRR | $5.1 \pm 0.2$ | $89 \pm 19$ | $1.0 \pm 0.1$ | $13.2 \pm 0.6$ | $8.0 \pm 0.2$ | $1.6 \pm 0.1$ | $1.8 \pm 0.1$ | $6.5 \pm 0.4$ | $1.1 \pm 0.1$ | $4.7 \pm 0.3$ | $9.9 \pm 0.4$ | $1.4 \pm 0.1$ |

The above Table 1 shows that the two methods using IW correction achieve the best and essentially equal performance. However, in terms of computational efficiency, our Nyström W-KRR method, offers significant time and memory savings. The number of Nyström points $m$ required by Nyström W-KRR is 1100, 1800, 1400, and 1550, for HAR70+, HARChildren, HHAR, and WISDM, respectively.

## 8 Conclusions and Future Work

In this paper, we showed that even under covariate shift, random projection techniques —particularly the Nyström method— can significantly enhance computational efficiency without any loss in learning performance. We provide new statistical bounds for our compressed Nyström algorithm, showing that it matches the optimal statistical guarantees of the full W-KRR model. Leveraging results from random projection theory, we developed novel technical proofs to account for the mismatch between training and test distributions and the potential unboundedness of the IW function. We evaluated the effectiveness of our approach through simulations and experiments on real-world datasets.

However, several questions remain open for future investigation. Although optimal rates are achieved in the well-specified case, the misalignment between the training and test distributions relative to the target function (see the *source condition* in Assumption 5) appears to play a critical role empirically, as it can make covariate shift either benign or severely adversarial. A deeper understanding of this phenomenon may come from a more detailed analysis of the constants in the learning bounds and their influence on the overall rate (see eq. (21) and the interaction between the covariance operators of source and target distributions).

## Acknowledgments

L. R. acknowledges the financial support of the European Commission (Horizon Europe grant ELIAS 101120237), the Ministry of Education, University and Research (FARE grant ML4IP R205T7J2KP), the European Research Council (grant SLING 819789), the U.S. Air Force Office of Scientific Research (FA8655-22-1-7034), the Ministry of Education, University and Research (grant BAC FAIR PE00000013, funded by the EU – NGEU), and MIUR (PRIN 202244A7YL). This work represents only the views of the authors. The European Commission and the other organizations are not responsible for any use that may be made of the information it contains. The research by Arnaud M. Watusadisi and L.R. was funded by a grant provided by the Liguria Region in collaboration with Leonardo SpA. The research by E.D.V has been partially supported by the MIUR grant PRIN 202244A7YL, by the PNRR project "Harmonic Analysis and Optimization in Infinite-Dimensional Statistical Learning - Future Artificial Intelligence Fair – Spoke 10" (CUP J33C24000410007) and by the MIUR Excellence Department Project awarded to Dipartimento di Matematica, Università di Genova (CUP D33C23001110001). E.D.V. is a member of the Gruppo Nazionale per l'Analisi Matematica, la Probabilità e le loro Applicazioni (GNAMPA) of the Istituto Nazionale di Alta Matematica (INdAM).

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

# A Derivation of the estimators

We define some quantities we will need when deriving our estimators. Following Section 2, we define the sampling operator $\widehat{Z}_m : \mathcal{H} \to \mathbb{R}^m$ associated with the subset $\{\widetilde{x}_1, \ldots, \widetilde{x}_m\} \subset \{x_1, \ldots, x_n\}$ as

$$\widehat{Z}_m : \mathcal{H}_m \to \mathbb{R}^m, \quad (\widehat{Z}_m f) = \langle f, K_{\widetilde{x}_i} \rangle_{\mathcal{H}_m}, \tag{24}$$

with its adjoint

$$\widehat{Z}_m^* : \mathbb{R}^m \to \mathcal{H}_m, \quad \widehat{Z}_m^* \widetilde{c} = \sum_{i=1}^m \widetilde{c}_i K_{\widetilde{x}_i}, \ \forall \widetilde{c} \in \mathbb{R}^m. \tag{25}$$

Moreover, consider the singular value decomposition (SVD) of $\widehat{Z}_m$ and $\widehat{Z}_m^*$

$$\widehat{Z}_m = UDV^*, \qquad \widehat{Z}_m^* = VDU^*$$

with $U : \mathbb{R}^k \to \mathbb{R}^m$, $D : \mathbb{R}^k \to \mathbb{R}^k$ the diagonal matrix of singular values sorted in non-decreasing order $D = \mathrm{diag}(\sigma_1, ..., \sigma_t)$ with $\sigma_1 \geqslant ... \geqslant \sigma_k > 0$, $V : \mathbb{R}^k \to \widetilde{\mathcal{H}}_m$, $k \leqslant m$ such that $U^*U = I_k$ and $V^*V = I_k$. The projection operator with range $\mathcal{H}_m$ is given by $P_m = VV^*$. We used this fact in Section 4. We derive an expression for the KRR minimizer when considering IW correction and Nyström approximation.

**Lemma 1** (Nyström W-KRR estimator)**.** *Given the minimization problem in* (12)*, the unique minimizer can be written as*

$$\widehat{f}_{\lambda,m}^w = V(V^*\widehat{S}^*\widehat{M}_w\widehat{S}V + \lambda\mathrm{I})^{-1}V^*\widehat{S}^*\widehat{M}_w\widehat{y} \tag{26}$$

*where $\lambda > 0$, $\widehat{y} = (y_1, ..., y_n)^T$ and the matrix $\widehat{M}_w = \mathrm{diag}(w(x_1), ..., w(x_n))$.*

*Proof.* Eq. (12) can be rewritten as

$$\left\| \widehat{M}_w^{\frac{1}{2}}(\widehat{S}P_m f - \widehat{y}) \right\|^2 = \left( \widehat{M}_w^{\frac{1}{2}}(\widehat{S}P_m f - \widehat{y}) \right)^T \left( \widehat{M}_w^{\frac{1}{2}}(\widehat{S}P_m f - \widehat{y}) \right). \tag{27}$$

Using first-order condition we have that

$$\frac{2}{n}P_m\widehat{S}^*\widehat{M}_w\widehat{S}P_m\widehat{f}_{\lambda,m}^w - \frac{2}{n}P_m\widehat{S}^*\widehat{M}_w\widehat{y} + 2\lambda\widehat{f}_{\lambda,m}^w = 0, \tag{28}$$

that is

$$(P_m\widehat{S}^*\widehat{M}_w\widehat{S}P_m + \lambda nI)\widehat{f}_{\lambda,m}^w = P_m\widehat{S}^*\widehat{M}_w\widehat{y}. \tag{29}$$

Replacing $P_m = VV^*$ we obtain

$$V(V^*\widehat{S}^*\widehat{M}_w\widehat{S}V + \lambda nI)V^*\widehat{f}_{\lambda,m}^w = VV^*\widehat{S}^*\widehat{M}_w\widehat{y}. \tag{30}$$

Multiplying both sides by $V^*$ and using that $(V^*\widehat{S}^*\widehat{M}_w\widehat{S}V + \lambda nI)$ is invertible, we obtain

$$V^*\widehat{f}_{\lambda,m}^w = (V^*\widehat{S}^*\widehat{M}_w\widehat{S}V + \lambda nI)^{-1}V^*\widehat{S}^*\widehat{M}_w\widehat{y}. \tag{31}$$

The result is obtained multiplying both side by $V$ and remembering that $\widehat{f}_{\lambda,m}^w \in \mathcal{H}_m$. $\qquad\square$

We can express our estimator in an alternative form which will be useful in the actual implementation of the algorithm.

**Lemma 2** (Nyström W-KRR estimator, representer theorem form)**.** *The above minimizer $\widehat{f}_{\lambda,m}^w$ can be also written as*

$$\widehat{f}_{\lambda,m}^w(x) = \sum_{i=1}^m \widetilde{c}_i^w K(\widetilde{x}_i, x), \quad \widetilde{c}^w = (\widehat{K}_{nm}^T\widehat{M}_w\widehat{K}_{nm} + n\lambda\widehat{K}_{mm})^{-1}\widehat{K}_{nm}^T\widehat{M}_w\widehat{y} \tag{32}$$

*where $\widehat{K}_{nm} = \widehat{S}\widehat{Z}_m^* \in \mathbb{R}^{n\times m}$, $(\widehat{K}_{nm})_{ij} = K(x_i, \widetilde{x}_j)$ and $\widehat{K}_{mm} = \widehat{Z}_m\widehat{Z}_m^* \in \mathbb{R}^{m\times m}$, $(\widehat{K}_{mm})_{ij} = K(\widetilde{x}_i, \widetilde{x}_j)$.*

*Proof.* Eq. (26) can be rewritten as

$$\begin{aligned}
\widehat{f}_{\lambda,m}^w &= V(V^*\widehat{S}^*\widehat{M}_w\widehat{S}V + \lambda nI)^{-1}V^*\widehat{S}^*\widehat{M}_w\widehat{y} \\
&= VDU^*UD^{-1}(V^*\widehat{S}^*\widehat{M}_w\widehat{S}V + \lambda nI)^{-1}D^{-1}U^*UDV^*\widehat{S}^*\widehat{M}_w\widehat{y} \\
&= \widehat{Z}_m^*(\widehat{Z}_m\widehat{S}^*\widehat{M}_w\widehat{S}\widehat{Z}_m^* + \lambda\widehat{Z}_m\widehat{Z}_m^*)^{\dagger}\widehat{Z}_m\widehat{S}^*\widehat{M}_w\widehat{y}
\end{aligned}$$

(33)

where we used $(FGH)^{\dagger} = H^{\dagger}G^{-1}F^{\dagger}$ (see the full-rank factorization of the pseudo-inverse (Ben-Israel & Greville, 2006)) with $F = UD$, $G = V^*\widehat{S}^*\widehat{M}_w\widehat{S}V + \lambda nI$ and $H = DU^T$.
Using the definitions of $\widehat{K}_{mm}$ and $\widehat{K}_{nm}$ we have

$$(\widehat{K}_{nm}^T\widehat{M}_w\widehat{K}_{nm} + \lambda n\widehat{K}_{mm})^{\dagger}\widehat{K}_{nm}^T\widehat{M}_wy = (\widehat{Z}_m\widehat{S}^*\widehat{M}_w\widehat{S}\widehat{Z}_m^* + \lambda\widehat{Z}_m\widehat{Z}_m^*)^{\dagger}\widehat{Z}_m\widehat{S}^*\widehat{M}_w\widehat{y}$$

(34)

and substituting this expression above we get the result. □

## B   Main proofs

To prove Theorem 1, we will need the following two propositions.

**Proposition 1** (Empirical Effective Dimension). *Let $\widehat{\mathcal{N}}_w(\lambda) = \mathrm{Tr}\,\widehat{\Sigma}_w\widehat{\Sigma}_{\lambda,w}^{-1}$. Under assumption 3 for any $\delta > 0$ and $\left(\frac{128(W+\sigma^2)\log^2(4/\delta)}{n}\right)^{\frac{1}{1+q}} \leqslant \lambda \leqslant \|\Sigma\|$, then the following holds with probability $1 - \delta$,*

$$\frac{|\widehat{\mathcal{N}}_w(\lambda) - \mathcal{N}_{\rho_X^{te}}(\lambda)|}{\mathcal{N}_{\rho_X^{te}}} \leqslant 2.$$

*Proof.* Let's call $\mathcal{N}_{\rho_X^{te}}(\lambda) = \mathcal{N}(\lambda)$ to simplify the notation. The proof partially follows the structure of Proposition 1 in Rudi et al. (2015), with some complications deriving from the presence of the (possibly unbounded) weights. Define $\widehat{B}^w = \Sigma_\lambda^{-1/2}(\Sigma - \widehat{\Sigma}_w)\Sigma_\lambda^{-1/2}$. Using Lemma 18 in Gogolashvili et al. (2023) we have that $\|\widehat{B}^w\|_{\mathrm{HS}} \leqslant 3/4$, when $n\lambda^{1+q} \geqslant 64\left(W+\sigma^2\right)\mathcal{N}_{\rho_X^{te}}(\lambda)^{1-q}\log^2\left(\frac{2}{\delta}\right)$. We can rewrite

$$\left|\widehat{\mathcal{N}}_w(\lambda) - \mathcal{N}(\lambda)\right| = \left|\mathrm{Tr}\left(\widehat{\Sigma}_{\lambda,w}^{-1}\widehat{\Sigma}_w - \Sigma\Sigma_\lambda^{-1}\right)\right| = \left|\lambda\,\mathrm{Tr}\,\widehat{\Sigma}_{\lambda,w}^{-1}\left(\widehat{\Sigma}_w - \Sigma\right)\Sigma_\lambda^{-1}\right|$$

(35)

$$= \left|\lambda\,\mathrm{Tr}\,\Sigma_\lambda^{-1/2}\left(\mathrm{I} - \widehat{B}^w\right)^{-1}\Sigma_\lambda^{-1/2}\left(\widehat{\Sigma}_w - \Sigma\right)\Sigma_\lambda^{-1/2}\Sigma_\lambda^{-1/2}\right|$$

(36)

$$= \left|\lambda\,\mathrm{Tr}\,\Sigma_\lambda^{-1/2}\left(\mathrm{I} - \widehat{B}^w\right)^{-1}\widehat{B}^w\Sigma_\lambda^{-1/2}\right|.$$

(37)

Following Rudi et al. (2015) and using that, for any symmetric linear operator $X : \mathcal{H} \to \mathcal{H}$ the following identity holds

$$(\mathrm{I} - X)^{-1}X = X + X(\mathrm{I} - X)^{-1}X.$$

Applying the above identity with $X = \widehat{B}$

$$\lambda\left|\mathrm{Tr}\,\Sigma_\lambda^{-1/2}\left(\mathrm{I} - \widehat{B}^w\right)^{-1}\widehat{B}^w\Sigma_\lambda^{-1/2}\right| \leqslant \lambda\underbrace{\left|\mathrm{Tr}\,\Sigma_\lambda^{-1/2}\widehat{B}^w\Sigma_\lambda^{-1/2}\right|}_{\mathbb{A}}$$

$$+ \lambda\underbrace{\left|\mathrm{Tr}\,\Sigma_\lambda^{-1/2}\widehat{B}^w\left(\mathrm{I} - \widehat{B}^w\right)^{-1}\widehat{B}^w\Sigma_\lambda^{-1/2}\right|}_{\mathbb{B}}.$$

To find an upper bound for $\mathbb{A}$ notice that

$$\mathbb{A} = \left|\mu - \frac{1}{n}\sum_{i=1}^n \xi_i\right|$$

with $\xi_i = \left\langle K_{x_i}, \lambda w(x_i)\Sigma_\lambda^{-2} K_{x_i} \right\rangle \in \mathbb{R}$ i.i.d. random variables with $i \in [n]$ and $\mu = \mathbb{E}[\xi_i]$. Using Lemma 18 in Gogolashvili et al. (2023) and a general version of Bernstein inequality requiring, instead of boundedness, only an appropriate control of moments (Boucheron et al., 2013), we have with probability greater than $1 - \delta$

$$\mathbb{A} \leqslant 4 \left( \frac{W}{\lambda n} + \sigma \sqrt{\frac{\mathcal{N}_{\rho_X^{te}}(\lambda)^{1-q}}{\lambda^{1+q} n}} \right) \log \left( \frac{2}{\delta} \right).$$

As regards $\mathbb{B}$, write $\mathbb{B} = \|Q\|_{HS}^2$ where $Q = \lambda^{1/2}\Sigma_\lambda^{-1/2}\widehat{B}^w \left( \mathrm{I} - \widehat{B}^w \right)^{-1/2}$, moreover

$$\|Q\|_{HS}^2 \leqslant \left\| \lambda^{1/2}\Sigma_\lambda^{-1/2} \right\|^2 \left\| \widehat{B}^w \right\|_{HS}^2 \left\| \left( \mathrm{I} - \widehat{B}^w \right)^{-1/2} \right\|^2 \leqslant 4\|\widehat{B}^w\|_{HS}^2,$$

since $\left\| \left( \mathrm{I} - \widehat{B}^w \right)^{-1/2} \right\|^2 \leqslant (1 - \|\widehat{B}^w\|)^{-1} \leqslant 4$, for $n\lambda^{1+q} \geqslant 64 \left( W + \sigma^2 \right) \mathcal{N}_{\rho_X^{te}}(\lambda)^{1-q} \log^2 \left( \frac{2}{\delta} \right)$.
Using again Lemma 18 in Gogolashvili et al. (2023) and a version of the Bernstein inequality for Hilbert space-valued random variables (see for example Caponnetto & De Vito (2007)) we obtain with probability greater than $1 - \delta$

$$\mathbb{B} \leqslant 16 \left( \frac{W}{\lambda n} + \sigma \sqrt{\frac{\mathcal{N}_{\rho_X^{te}}(\lambda)^{1-q}}{\lambda^{1+q} n}} \right)^2 \log^2 \left( \frac{2}{\delta} \right)$$

Putting all together, with probability $1 - \delta$:

$$\left| \widehat{\mathcal{N}}_w(\lambda) - \mathcal{N}(\lambda) \right| \leqslant 4 \left( \frac{W}{\lambda n} + \sigma \sqrt{\frac{\mathcal{N}_{\rho_X^{te}}(\lambda)^{1-q}}{\lambda^{1+q} n}} \right) \log \left( \frac{4}{\delta} \right) + 16 \left( \frac{W}{\lambda n} + \sigma \sqrt{\frac{\mathcal{N}_{\rho_X^{te}}(\lambda)^{1-q}}{\lambda^{1+q} n}} \right)^2 \log^2 \left( \frac{4}{\delta} \right)$$
(38)

$$\leqslant 4 \left( \frac{W}{\lambda n} + \sigma \sqrt{\frac{1}{\lambda^{\gamma(1-q)+1+q} n}} \right) \log \left( \frac{4}{\delta} \right) + 16 \left( \frac{W}{\lambda n} + \sigma \sqrt{\frac{1}{\lambda^{\gamma(1-q)+1+q} n}} \right)^2 \log^2 \left( \frac{4}{\delta} \right)$$
(39)

Using that $\mathcal{N}(\lambda) \geqslant \|\Sigma\Sigma_\lambda^{-1}\| \geqslant 1/2$ if $\lambda \leqslant \|\Sigma\|$ we have

$$\left| \widehat{\mathcal{N}}_w(\lambda) - \mathcal{N}(\lambda) \right| \tag{40}$$

$$\leqslant \left( 4 \left( \frac{W}{\mathcal{N}_{\rho_X^{te}}(\lambda)\lambda n} + \sigma \sqrt{\frac{1}{\mathcal{N}_{\rho_X^{te}}(\lambda)^{1+q}\lambda^{1+q} n}} \right) \log \left( \frac{4}{\delta} \right) + \right. \tag{41}$$

$$\left. + 16 \left( \frac{W}{\mathcal{N}_{\rho_X^{te}}(\lambda)\lambda n} + \sigma \sqrt{\frac{1}{\mathcal{N}_{\rho_X^{te}}(\lambda)^{1+q}\lambda^{1+q} n}} \right)^2 \log^2 \left( \frac{4}{\delta} \right) \right) \mathcal{N}_{\rho_X^{te}}(\lambda) \tag{42}$$

$$\leqslant \left( 4 \left( \frac{2W}{\lambda n} + \sigma \sqrt{\frac{4}{\lambda^{1+q} n}} \right) \log \left( \frac{4}{\delta} \right) + 16 \left( \frac{2W}{\lambda n} + \sigma \sqrt{\frac{4}{\lambda^{1+q} n}} \right)^2 \log^2 \left( \frac{4}{\delta} \right) \right) \mathcal{N}_{\rho_X^{te}}(\lambda). \tag{43}$$

Then for $\left( \frac{256(W+\sigma^2)\log^2(4/\delta)}{n} \right)^{\frac{1}{1+q}} \leqslant \lambda \leqslant \|\Sigma\|$ with probability $1 - \delta$

$$\left| \widehat{\mathcal{N}}_w(\lambda) - \mathcal{N}(\lambda) \right| \leqslant 2\mathcal{N}(\lambda)$$

$\square$

**Proposition 2** (Nyström approximation with ALS sampling). *Let $\left( \hat{l}_i(t) \right)_{i=1}^n$ be the collection of approximate leverage scores. Let $\lambda > 0$ and $P_\lambda$ be defined as $P_\lambda(i) = \hat{l}_i(\lambda) / \sum_{j \in N} \hat{l}_j(\lambda)$ for any*

$i \in N$ with $N = \{1, \ldots, n\}$. Let $\mathfrak{I} = (i_1, \ldots, i_m)$ be a collection of indices independently sampled with replacement from $N$ according to the probability distribution $P_\lambda$. Let $P_m$ be the projection operator on the subspace $\mathcal{H}_m = \mathrm{span}\left\{ K_{x_j} \mid j \in J \right\}$ and $J$ be the subcollection of $\mathfrak{I}$ with all the duplicates removed. Under Assumption 3 and 6, for any $\delta > 0$ the following holds with probability $1 - 2\delta$

$$\|(I - P_m)\Sigma_\lambda^{1/2}\| \leqslant 3\lambda$$

when the following conditions are satisfied:

- there exists $T \geqslant 1$ and $\lambda_0 > 0$ such that $\left( \hat{l}_i(t) \right)_{i=1}^n$ are $T$-approximate leverage scores for any $t \geqslant \lambda_0$ (see Definition 1),

- $\lambda_0 \vee \left( \frac{256(W+\sigma^2)\log^2(4/\delta)}{n} \right)^{\frac{1}{\gamma(1-q)+1+q}} \leqslant \lambda \leqslant \|\Sigma\|$,

- $m \geqslant 144T^2 \mathcal{N}_{\rho_X^{te}}(\lambda) \log \frac{8n}{\delta}$.

*Proof.* Let's call $\Sigma = \Sigma_{\rho_X^{te}}$ to simplify the notation. Define $\tau = \delta/4$. Next, define the diagonal matrix $H \in \mathbb{R}^{n \times n}$ with $(H)_{ii} = 0$ when $P_\lambda(i) = 0$ and $(H)_{ii} = \frac{nq(i)}{mP_\lambda(i)}$ when $P_\lambda(i) > 0$, where $q(i)$ is the number of times the index $i$ is present in the collection $\mathfrak{I}$. We have that

$$\widehat{S}_w^* H \widehat{S}_w = \frac{1}{m} \sum_{i=1}^n w(x_i) \frac{q(i)}{P_\lambda(i)} K_{x_i} \otimes K_{x_i} = \frac{1}{m} \sum_{j \in J} w(x_j) \frac{q(j)}{P_\lambda(j)} K_{x_j} \otimes K_{x_j}.$$

Now, considering that $\frac{q(j)}{P_\lambda(j)} > 0$ for any $j \in J$, thus $\mathrm{ran}\, \widehat{S}_w^* H \widehat{S}_w = \mathcal{H}_m$. Therefore, by using Prop. 3 and 7 in Rudi et al. (2015), we exploit the fact that the range of $P_m$ is the same of $\widehat{S}_w^* H \widehat{S}_w$, to obtain

$$\left\| (I - P_m) \Sigma_\lambda^{1/2} \right\|^2 \leqslant \lambda \left\| \left( \widehat{S}_w^* H \widehat{S}_w + \lambda I \right)^{-1/2} \Sigma^{1/2} \right\|^2 \leqslant \frac{\lambda}{1 - \beta(\lambda)},$$

with $\beta(\lambda) = \lambda_{\max} \left( \Sigma_\lambda^{-1/2} \left( \Sigma - \widehat{S}_w^* H \widehat{S}_w \right) \Sigma_\lambda^{-1/2} \right)$. Considering that the function $(1 - x)^{-1}$ is increasing on $-\infty < x < 1$, in order to bound $\lambda/(1 - \beta(\lambda))$ we need an upperbound for $\beta(\lambda)$. Here we split $\beta(\lambda)$ in the following way,

$$\beta(\lambda) \leqslant \underbrace{\lambda_{\max} \left( \Sigma_\lambda^{-1/2} \left( \Sigma - \widehat{\Sigma}_w \right) \Sigma_\lambda^{-1/2} \right)}_{\beta_1(\lambda)} + \underbrace{\lambda_{\max} \left( \Sigma_\lambda^{-1/2} \left( \widehat{\Sigma}_w - \widehat{S}_w^* H \widehat{S}_w \right) \Sigma_\lambda^{-1/2} \right)}_{\beta_2(\lambda)}.$$

$\beta_1$ can be bounded as in eq. (44).
As regards $\beta_2$:

$$\beta_2(\lambda) \leqslant \left\| \Sigma_\lambda^{-1/2} \left( \widehat{\Sigma}_w - \widehat{S}_w^* H \widehat{S}_w \right) \Sigma_\lambda^{-1/2} \right\|$$

$$\leqslant \left\| \Sigma_\lambda^{-1/2} \widehat{\Sigma}_{w\lambda}^{1/2} \right\|^2 \left\| \widehat{\Sigma}_{w\lambda}^{-1/2} \left( \widehat{\Sigma}_w - \widehat{S}_w^* H \widehat{S}_w \right) \widehat{\Sigma}_{w\lambda}^{-1/2} \right\|.$$

Let

$$\beta_3(\lambda) = \left\| \widehat{\Sigma}_{w\lambda}^{-1/2} \left( \widehat{\Sigma}_w - \widehat{S}_w^* H \widehat{S}_w \right) \widehat{\Sigma}_{w\lambda}^{-1/2} \right\| = \left\| \widehat{\Sigma}_{w\lambda}^{-1/2} \widehat{S}_w^* (I - H) \widehat{S}_w \widehat{\Sigma}_{w\lambda}^{-1/2} \right\|.$$

Note that $\widehat{S}_w \widehat{\Sigma}_{w\lambda}^{-1} \widehat{S}_w^* = \widehat{S}_w (\widehat{S}_w^* \widehat{S}_w + \lambda \mathrm{I})^{-1} \widehat{S}_w^* = \left( \widehat{K}_w + \lambda n I \right)^{-1} \widehat{K}_w$ since $\widehat{K}_w = n \widehat{S}_w \widehat{S}_w^*$, with $(\widehat{K}_w)_{ij} = w(x_i)^{1/2} w(x_j)^{1/2} K(x_i, x_j)$.

Thus, if we let $UDU^\top$ be the eigendecomposition of $\widehat{K}_w$, we have that $\left( \widehat{K}_w + \lambda n I \right)^{-1} \widehat{K}_w = U(D + \lambda n I)^{-1} D U^\top$ and thus $\widehat{S}_w \widehat{\Sigma}_{w\lambda}^{-1} \widehat{S}_w^* = U(D + \lambda n I)^{-1} D U^\top$. In particular this implies that $\widehat{S}_w \widehat{\Sigma}_{w\lambda}^{-1} \widehat{S}_w^* = U \widehat{Q}^{1/2} \widehat{Q}^{1/2} U^\top$ with $\widehat{Q} = (D + \lambda n I)^{-1} D$. Therefore we have

$$\beta_3(\lambda) = \left\| \widehat{\Sigma}_{w\lambda}^{-1/2} \widehat{S}_w^* (I - H) \widehat{S}_w \widehat{\Sigma}_{w\lambda}^{-1/2} \right\| = \left\| \widehat{Q}^{1/2} U^\top (I - H) U \widehat{Q}^{1/2} \right\|.$$

Consider the matrix $A = \widehat{Q}^{1/2}U^\top$ and let $a_i$ be the $i$-th column of $A$, and $e_i$ be the $i$-th canonical basis vector for each $i \in N$. We prove that $\|a_i\|^2 = l_i(\lambda)$, the true leverage score, since

$$\|a_i\|^2 = \left\|\widehat{Q}^{1/2}U^\top e_i\right\|^2 = e_i^\top U\widehat{Q}U^\top e_i = \left(\left(\widehat{K}_w + \lambda n I\right)^{-1}\widehat{K}_w\right)_{ii} = l_i(\lambda)$$

Considering that $\sum_{k=1}^n \frac{q(k)}{P_\lambda(k)}a_k a_k^\top = \sum_{i=\Im}\frac{1}{P_\lambda(i)}a_i a_i^\top$, we have

$$\beta_3(\lambda) = \left\|AA^\top - \frac{1}{m}\sum_{i\in\Im}\frac{1}{P_\lambda(i)}a_i a_i^\top\right\|.$$

Moreover, by the $T$-approximation property of the approximate leverage scores (see Def. 1 in Rudi et al. (2015)), we have that for all $i \in \{1,\dots,n\}$, when $\lambda \geqslant \lambda_0$, the following holds with probability $1 - \delta$

$$P_\lambda(i) = \frac{\hat{l}_i(\lambda)}{\sum_j \hat{l}_j(\lambda)} \geqslant T^{-2}\frac{l_i(\lambda)}{\sum_j l_j(\lambda)} = T^{-2}\frac{\|a_i\|^2}{\operatorname{Tr} AA^\top}.$$

Then, we can apply Prop. 9 in Rudi et al. (2015), so that, after a union bound, we obtain the following inequality with probability $1 - \delta - \tau$:

$$\beta_3(\lambda) \leqslant \frac{2\|A\|^2 \log\frac{2n}{\tau}}{3m} + \sqrt{\frac{2\|A\|^2 T^2 \operatorname{Tr} AA^\top \log\frac{2n}{\tau}}{m}} \leqslant \frac{2\log\frac{2n}{\tau}}{3m} + \sqrt{\frac{2T^2\widehat{\mathcal{N}}_w(\lambda)\log\frac{2n}{\tau}}{m}},$$

where the last step follows from $\|A\|^2 = \left\|\left(\widehat{K}_w + \lambda n I\right)^{-1}\widehat{K}_w\right\| \leqslant 1$ and $\operatorname{Tr}\left(AA^\top\right) = \operatorname{Tr}\left(\widehat{\Sigma}_{w\lambda}^{-1}\widehat{\Sigma}_w\right) := \widehat{\mathcal{N}}_w(\lambda)$. Applying proposition 1 we have that $\widehat{\mathcal{N}}_w(\lambda) \leqslant 3\mathcal{N}_{\rho_X^{te}}(\lambda)$ with probability $1 - \tau$, when $\left(\frac{128(W+\sigma^2)\log^2(4/\delta)}{n}\right)^{\frac{1}{1+q}} \leqslant \lambda \leqslant \|\Sigma\|$. Thus, by taking a union bound again, we have

$$\beta_3(\lambda) \leqslant \frac{2\log\frac{2n}{\tau}}{3m} + \sqrt{\frac{16T^2\mathcal{N}_{\rho_X^{te}}(\lambda)\log\frac{2n}{\tau}}{m}}$$

with probability $1 - 2\tau - \delta$. The last step is to bound $\left\|\Sigma_\lambda^{-1/2}\widehat{\Sigma}_{w\lambda}^{1/2}\right\|^2$, as follows

$$\left\|\Sigma_\lambda^{-1/2}\widehat{\Sigma}_{\lambda w}^{1/2}\right\|^2 = \left\|\Sigma_\lambda^{-1/2}\widehat{\Sigma}_{\lambda w}\Sigma_\lambda^{-1/2}\right\| = \left\|I + \Sigma_\lambda^{-1/2}\left(\widehat{\Sigma}_w - \Sigma\right)\Sigma_\lambda^{-1/2}\right\| \leqslant 1 + \eta,$$

with $\eta = \left\|\Sigma_\lambda^{-1/2}\left(\widehat{\Sigma}_w - \Sigma\right)\Sigma_\lambda^{-1/2}\right\|$. We can bound $\eta$ using Lemma 18 in Gogolashvili et al. (2023) (see eq. (44)). Finally, by collecting the above results and taking a union bound we have

$$\beta(\lambda) \leqslant 4\left(\frac{W}{\lambda n} + \sqrt{\frac{\sigma^2\mathcal{N}_{\rho_X^{te}}(\lambda)^{1-q}}{\lambda^{1+q}n}}\right)\log\left(\frac{2}{\tau}\right) + (1+\eta)\left(\frac{2\log\frac{2n}{\tau}}{3m} + \sqrt{\frac{16T^2\mathcal{N}_{\rho_X^{te}}(\lambda)\log\frac{2n}{\tau}}{m}}\right),$$

with probability $1 - 4\tau - \delta = 1 - 2\delta$ when $\left(\frac{256(W+\sigma^2)\log^2(4/\delta)}{n}\right)^{\frac{1}{1+q}} \leqslant \lambda \leqslant \|\Sigma\|$. Note that, if we select $\left(\frac{256(W+\sigma^2)\log^2(4/\delta)}{n}\right)^{\frac{1}{\gamma(1-q)+1+q}} \leqslant \lambda \leqslant \|\Sigma\|$, and $m \geqslant 144T^2\mathcal{N}_{\rho_X^{te}}(\lambda)\log\frac{8n}{\delta}$, we have $\beta(\lambda) \leqslant 5/6$, so that

$$\left\|\left(I - P_m\right)\Sigma^{1/2}\right\|^2 \leqslant 6\lambda$$

with probability $1 - 2\delta$. $\qquad\square$

We can now proceed with the proof of Theorem 1.

**Proof of Theorem 1**

*proof of Theorem 1.* We split the excess risk as

$$
\left| \mathcal{R}(\widehat{f}_{\lambda,m}^w) - \mathcal{R}(f_{\mathcal{H}}) \right|^{1/2} = \left\| \widehat{f}_{\lambda,m}^w - f_{\mathcal{H}} \right\|_{\rho_X^{te}} = \left\| \Sigma^{1/2}(\widehat{f}_{\lambda,m}^w - f_{\mathcal{H}}) \right\|_{\mathcal{H}}
$$

$$
= \left\| \Sigma^{1/2}(V(V^*\widehat{\Sigma}_w V + \lambda \mathrm{I})^{-1}V^*\widehat{S}^* M_w y - f_{\mathcal{H}}) \right\|_{\mathcal{H}}
$$

$$
\leqslant \underbrace{\left\| \Sigma^{1/2}V(V^*\widehat{\Sigma}_w V + \lambda \mathrm{I})^{-1}V^*\widehat{S}^* M_w(y - \widehat{S}f_{\mathcal{H}}) \right\|_{\mathcal{H}}}_{\mathbb{A}} +
$$

$$
+ \underbrace{\left\| \Sigma^{1/2}(\mathrm{I} - V(V^*\widehat{\Sigma}_w V + \lambda \mathrm{I})^{-1}V^*\widehat{\Sigma}_w)f_{\mathcal{H}} \right\|_{\mathcal{H}}}_{\mathbb{B}}
$$

**Term $\mathbb{A}$**

$$
\mathbb{A} \leqslant \underbrace{\left\| \Sigma^{1/2}\widehat{\Sigma}_{w\lambda}^{-1/2} \right\|}_{\mathbb{A}_1} \underbrace{\left\| \widehat{\Sigma}_{w\lambda}^{1/2}V(V^*\widehat{\Sigma}_{w\lambda}V)^{-1}V^*\widehat{\Sigma}_{w\lambda}^{1/2} \right\|}_{\mathbb{A}_2} \underbrace{\left\| \widehat{\Sigma}_{w\lambda}^{-1/2}\Sigma_{\lambda}^{1/2} \right\|}_{\mathbb{A}_3 = \beta} \underbrace{\left\| \Sigma_{\lambda}^{-1/2}\widehat{S}^* M_w(y - \widehat{S}f_{\mathcal{H}}) \right\|_{\mathcal{H}}}_{\mathbb{A}_4}
$$

- $\mathbb{A}_1$:

$$
\mathbb{A}_1 \leqslant \underbrace{\left\| \Sigma^{1/2}\Sigma_{\lambda}^{-1/2} \right\|}_{\leqslant 1} \underbrace{\left\| \Sigma_{\lambda}^{1/2}\widehat{\Sigma}_{w\lambda}^{-1/2} \right\|}_{\beta} \leqslant \beta
$$

- $\mathbb{A}_2$: using lemma 8 in Rudi et al. (2015) it's easy to show that $\left\| \widehat{\Sigma}_{w\lambda}^{1/2}V(V^*\widehat{\Sigma}_{w\lambda}V)^{-1}V^*\widehat{\Sigma}_{w\lambda}^{1/2} \right\|^2 = \left\| \widehat{\Sigma}_{w\lambda}^{1/2}V(V^*\widehat{\Sigma}_{w\lambda}V)^{-1}V^*\widehat{\Sigma}_{w\lambda}^{1/2} \right\|$, and therefore the only possible values for $\mathbb{A}_2$ are 0 and 1. Then

$$
\mathbb{A}_2 \leqslant 1
$$

- $\mathbb{A}_3 = \beta$: using proposition 7 in Rudi et al. (2015) we have

$$
\beta \leqslant \frac{1}{1 - b}
$$

if $b = \lambda_{\max}\left[ \Sigma_{\lambda}^{-1/2}(\Sigma - \widehat{\Sigma}_w)\Sigma_{\lambda}^{-1/2} \right] < 1$.

Applying Lemma 18 in Gogolashvili et al. (2023) we get, with probability greater than $1 - \delta$

$$
\lambda_{\max}\left[ \Sigma_{\lambda}^{-1/2}(\Sigma - \widehat{\Sigma}_w)\Sigma_{\lambda}^{-1/2} \right] \leqslant \left\| \Sigma_{\lambda}^{-\frac{1}{2}}\left( \Sigma - \widehat{\Sigma}_w \right)\Sigma_{\lambda}^{-\frac{1}{2}} \right\|_{\mathrm{HS}} \leqslant 4\left( \frac{W}{\lambda n} + \sqrt{\frac{\sigma^2 \mathcal{N}_{\rho_X^{te}}(\lambda)^{1-q}}{\lambda^{1+q}n}} \right)\log\left( \frac{2}{\delta} \right) \tag{44}
$$

and for $n\lambda^{1+q} \geqslant 64\left( W + \sigma^2 \right)\mathcal{N}_{\rho_X^{te}}(\lambda)^{1-q}\log^2\left( \frac{2}{\delta} \right)$, with probability greater than $1 - \delta$ the above quantity is less or equal than $3/4$.

- $\mathbb{A}_4$:

To control this term we use lemma 19 in Gogolashvili et al. (2023), where $\xi_i = \Sigma_{\lambda}^{-1/2}w(x_i)K_{x_i}y_i$. We obtain that, with probability greater or equal than $1 - \delta$

$$
\left\| \frac{1}{n}\sum^n \xi_i - \mathbb{E}[\xi] \right\|_{\mathcal{H}} = \left\| \Sigma_{\lambda}^{-1/2}\widehat{S}^* M_w(y - \widehat{S}f_{\mathcal{H}}) \right\|_{\mathcal{H}} \leqslant 4B\left( \frac{W}{n\sqrt{\lambda}} + \sqrt{\frac{\sigma^2 \mathcal{N}_{\rho_X^{te}}(\lambda)^{1-q}}{n\lambda^q}} \right)\log\left( \frac{2}{\delta} \right).
$$

**Term** $\mathbb{B}$

As regards $\mathbb{B}$, following Rudi et al. (2015) we proceed as follows. Noting that $V(V^*\widehat{\Sigma}_{w\lambda}V)^{-1}V^*\widehat{\Sigma}_{w\lambda}VV^* = VV^*$, we have

$$I-V(V^*\widehat{\Sigma}_{w\lambda}V)^{-1}V^*\widehat{\Sigma}_w = I - V(V^*\widehat{\Sigma}_{w\lambda}V)^{-1}V^*\widehat{\Sigma}_{w\lambda} + \lambda V(V^*\widehat{\Sigma}_{w\lambda}V)^{-1}V^*$$
$$= I - V(V^*\widehat{\Sigma}_{w\lambda}V)^{-1}V^*\widehat{\Sigma}_{w\lambda}VV^* - V(V^*\widehat{\Sigma}_{w\lambda}V)^{-1}V^*\widehat{\Sigma}_{w\lambda}\left(I - VV^*\right) + \lambda V(V^*\widehat{\Sigma}_{w\lambda}V)^{-1}V^*$$
$$= (I - VV^*) + \lambda V(V^*\widehat{\Sigma}_{w\lambda}V)^{-1}V^* - V(V^*\widehat{\Sigma}_{w\lambda}V)^{-1}V^*\widehat{\Sigma}_{w\lambda}\left(I - VV^*\right).$$

By assumption 5, we have $\left\|\Sigma_{w\lambda}^{-(r-1/2)}f_{\mathcal{H}}\right\|_{\mathcal{H}} \leqslant \left\|\Sigma_{w\lambda}^{-r}f_{\mathcal{H}}\right\|_{\mathcal{H}} \leqslant \left\|\Sigma_w^{-r}f_{\mathcal{H}}\right\|_{\mathcal{H}} \leqslant R$. Define $r' := r - 1/2$ to simplify the notation. Using the above decomposition, we can rewrite term $\mathbb{B}$ as

$$\mathbb{B} \leqslant \left\|\Sigma^{1/2}\left(I - V(V^*\widehat{\Sigma}_{w\lambda}V)^{-1}V^*\widehat{\Sigma}_w\right)\Sigma_{w\lambda}^{r'}\right\|\left\|\Sigma_{w\lambda}^{-r'}f_{\mathcal{H}}\right\|_{\mathcal{H}}$$

$$\leqslant R\left\|\Sigma^{1/2}\Sigma_\lambda^{-1/2}\right\|\left\|\Sigma^{1/2}\left(I - VV^*\right)\Sigma_\lambda^{r'}\right\| + R\lambda\left\|\Sigma^{1/2}\widehat{\Sigma}_{w\lambda}^{-1/2}\right\|\left\|\widehat{\Sigma}_{w\lambda}^{1/2}V(V^*\widehat{\Sigma}_{w\lambda}V)^{-1}V^*\Sigma_\lambda^{r'}\right\|$$

$$+ R\left\|\Sigma^{1/2}\widehat{\Sigma}_{w\lambda}^{-1/2}\right\|\left\|\widehat{\Sigma}_{w\lambda}^{1/2}V(V^*\widehat{\Sigma}_{w\lambda}V)^{-1}V^*\widehat{\Sigma}_{w\lambda}^{1/2}\right\|\left\|\widehat{\Sigma}_{w\lambda}^{1/2}\Sigma_\lambda^{-1/2}\right\|\left\|\Sigma_\lambda^{1/2}\left(I - VV^*\right)\Sigma_\lambda^{r'}\right\|$$

$$\leqslant R(1+\beta\theta)\underbrace{\left\|\Sigma_\lambda^{1/2}\left(I - VV^*\right)\Sigma_\lambda^{r'}\right\|}_{\mathbb{B}.1} + R\beta\,\lambda\underbrace{\left\|\widehat{\Sigma}_{w\lambda}^{1/2}V(V^*\widehat{\Sigma}_{w\lambda}V)^{-1}V^*\Sigma_\lambda^{r'}\right\|}_{\mathbb{B}.2},$$

with $\theta = \left\|\widehat{\Sigma}_{w\lambda}^{1/2}\Sigma_\lambda^{-1/2}\right\|$.

- $\mathbb{B}.1$:

$$\mathbb{B}.1 = \left\|\Sigma_\lambda^{1/2}\left(I - VV^*\right)^2\Sigma_\lambda^{r'}\right\| \leqslant \left\|\Sigma_\lambda^{1/2}\left(I - VV^*\right)\right\|\left\|\left(I - VV^*\right)\Sigma_\lambda^{r'}\right\|.$$

  Since $VV^*$ is a projection operator, we have that $(I - VV^*) = (I - VV^*)^s$, for any $s > 0$, therefore, by applying Cordes inequality (see proposition 6) to $\left\|\left(I - VV^*\right)\Sigma_\lambda^{r'}\right\|$, we have

$$\left\|\left(\mathrm{I} - VV^*\right)\Sigma_\lambda^{r'}\right\| = \left\|\left(\mathrm{I} - VV^*\right)^{2r}\Sigma_\lambda^{\frac{1}{2}2r'}\right\| \leqslant \left\|\left(\mathrm{I} - VV^*\right)\Sigma_\lambda^{1/2}\right\|^{2r'}.$$

  This term can be controlled using Proposition 2 above.

- $\mathbb{B}.2$:

$$\mathbb{B}.2 \leqslant \lambda\left\|\widehat{\Sigma}_{w\lambda}^{1/2}V(V^*\widehat{\Sigma}_{w\lambda}V)^{-1}V^*\widehat{\Sigma}_{w\lambda}^{r'}\right\|\left\|\widehat{\Sigma}_{w\lambda}^{-r'}\Sigma_\lambda^{r'}\right\|$$

$$\leqslant \lambda\left\|\widehat{\Sigma}_{w\lambda}^{1/2}V(V^*\widehat{\Sigma}_{w\lambda}V)^{-1}V^*\widehat{\Sigma}_{w\lambda}^{r'}\right\|\left\|\widehat{\Sigma}_{w\lambda}^{-1/2}\Sigma_\lambda^{1/2}\right\|^{2r'}$$

$$\leqslant \beta^{2r'}\lambda\left\|\left(V^*\widehat{\Sigma}_{w\lambda}V\right)^{1/2}\left(V^*\widehat{\Sigma}_{w\lambda}V\right)^{-1}\left(V^*\widehat{\Sigma}_{w\lambda}V\right)^{r'}\right\|$$

$$= \beta^{2r'}\lambda\left\|\left(V^*\widehat{\Sigma}_wV + \lambda I\right)^{-(1/2-r')}\right\| \leqslant \beta\lambda^{1/2+r'} = \beta\lambda^r,$$

  where the first step is obtained multiplying and dividing by $\widehat{\Sigma}_{w\lambda}^{r'}$, the second step by applying Cordes inequality, the third step by Prop. 6 in Rudi et al. (2015).

Putting all together, for $\left(\frac{256(W+\sigma^2)\log^2(4/\delta)}{n}\right)^{\frac{1}{\gamma(1-q)+1+q}} \leqslant \lambda \leqslant \|\Sigma\|$, and $m \geqslant 144T^2Q\lambda^{-\gamma}\log\frac{8n}{\delta}$, with probability greater or equal than $1-\delta$

$$\left|\mathcal{R}(\widehat{f}_{\lambda,m}^w) - \mathcal{R}(f_{\mathcal{H}})\right|^{1/2} \leqslant 4B\beta^2\left(\frac{W}{n\sqrt{\lambda}} + \sqrt{\frac{\sigma^2}{n\lambda^{\gamma(1-q)+q}}}\right)\log\left(\frac{8}{\delta}\right) + 3R(1+\beta\theta)\lambda^r + R\beta^2\lambda^r.$$

Under the above conditions on $\lambda$ we have that $\beta \leqslant 4$ and $\theta \leqslant 2$ and then for $\left(\frac{256(W+\sigma^2)\log^2(4/\delta)}{n}\right)^{\frac{1}{\gamma(1-q)+1+q}} \leqslant \lambda \leqslant \|\Sigma\|$, and $m \geqslant 144T^2Q\lambda^{-\gamma}\log\frac{8n}{\delta}$, with probability greater or equal than $1-\delta$

$$\left|\mathcal{R}(\widehat{f}^w_{\lambda,m}) - \mathcal{R}(f_\mathcal{H})\right|^{1/2} \leqslant 64B\left(\frac{W}{n\sqrt{\lambda}} + \sqrt{\frac{\sigma^2}{n\lambda^{\gamma(1-q)+q}}}\right)\log\left(\frac{8}{\delta}\right) + 43R\lambda^r.$$

$\square$

## C Known results

In this section, we derive tight bounds for the effective dimension $\mathcal{N}_\rho(\lambda)$ defined in Definition 2 when assuming polynomial decay of the eigenvalues $\sigma_j(\Sigma)$ of the covariance operator $\Sigma$.

**Proposition 3** (Polynomial eigenvalues decay, Proposition 3 in Caponnetto & De Vito (2007)). *If for some $\gamma \in \mathbb{R}^+$ and $1 < \beta < +\infty$*

$$\sigma_i \leqslant \gamma i^{-\beta}$$

*then*

$$\mathcal{N}_\rho(\lambda) \leqslant \gamma\frac{\beta}{\beta - 1}\lambda^{-1/\beta} \tag{45}$$

*Proof.* Since the function $\sigma/(\sigma + \lambda)$ is increasing in $\sigma$ and using the spectral theorem $\Sigma = UDU^*$ combined with the fact that $\mathrm{Tr}(UDU^*) = \mathrm{Tr}(U(U^*D)) = \mathrm{Tr}D$

$$\mathcal{N}_\rho(\lambda) = \mathrm{Tr}(\Sigma(\Sigma + \lambda I)^{-1}) = \sum_{i=1}^{\infty}\frac{\sigma_i}{\sigma_i + \lambda} \leqslant \sum_{i=1}^{\infty}\frac{\gamma}{\gamma + i^\beta\lambda} \tag{46}$$

The function $\gamma/(\gamma + x^\beta\lambda)$ is positive and decreasing, so

$$\begin{aligned}
\mathcal{N}_\rho(\lambda) &\leqslant \int_0^\infty \frac{\gamma}{\gamma + x^\beta\lambda}dx \\
&= \alpha^{-1/\beta}\int_0^\infty \frac{\gamma}{\gamma + \tau^\beta}d\tau \\
&\leqslant \gamma\frac{\beta}{\beta - 1}\lambda^{-1/\beta}
\end{aligned} \tag{47}$$

since $\int_0^\infty (\gamma + \tau^\beta)^{-1} \leqslant \beta/(\beta - 1)$. $\square$

Further improvements can be found assuming exponential decay of the eigenvalues $\sigma_j(\Sigma)$ of $\Sigma$.

**Proposition 4** (Exponential eigenvalues decay, Proposition 5 in Della Vecchia et al. (2021)). *If for some $\gamma, \beta \in \mathbb{R}^+$ $\sigma_i \leqslant \gamma e^{-\beta i}$ then*

$$\mathcal{N}_\rho(\lambda) \leqslant \frac{\log(1 + \gamma/\lambda)}{\beta}. \tag{48}$$

*Proof.*

$$\mathcal{N}_\rho(\lambda) = \sum_{i=1}^{\infty}\frac{\sigma_i}{\sigma_i + \lambda} = \sum_{i=1}^{\infty}\frac{1}{1 + \lambda/\sigma_i} \leqslant \sum_{i=1}^{\infty}\frac{1}{1 + \lambda'e^{\beta i}} \leqslant \int_0^{+\infty}\frac{1}{1 + \lambda'e^{\beta x}}dx, \tag{49}$$

where $\lambda' = \lambda/\gamma$. Using the change of variables $t = e^{\beta x}$ we get

$$\begin{aligned}
(49) &= \frac{1}{\beta}\int_1^{+\infty}\frac{1}{1 + \lambda't}\frac{1}{t}dt = \frac{1}{\beta}\int_1^{+\infty}\left[\frac{1}{t} - \frac{\lambda'}{1 + \lambda't}\right]dt = \frac{1}{\beta}\left[\log t - \log(1 + \lambda't)\right]_1^{+\infty} \\
&= \frac{1}{\beta}\left[\log\left(\frac{t}{1 + \lambda't}\right)\right]_1^{+\infty} = \frac{1}{\beta}\left[\log(1/\lambda') + \log(1 + \lambda')\right].
\end{aligned} \tag{50}$$

So we finally obtain

$$\mathcal{N}_\rho(\lambda) \leqslant \frac{1}{\beta}\Big[\log(\gamma/\lambda) + \log(1 + \lambda/\gamma)\Big] = \frac{\log(1 + \gamma/\lambda)}{\beta}. \tag{51}$$

$\square$

Next proposition gives some relations between weights and covariance operators.

**Proposition 5.** *Suppose that $\rho_X^{te}$ is absolutely continuous with respect to $\rho_X^v$, and the Radon-Nikodym derivative $d\rho_X^{te}/d\rho_X^v$ is bounded by $G \in \mathbb{R}$. Then*

$$\big\|\Sigma\Sigma_{v\lambda}^{-1}\big\| \leqslant G,$$

*see Gogolashvili et al. (2023).*
*Moreover, if $w(x) < \infty$ and $v(x) > 0$ for all $x \in \mathcal{X}$, with $w$ and $v$ defined as in eq. (3) and (18) respectively, then*

$$\big\|\Sigma\Sigma_{v\lambda}^{-1}\big\| \leqslant \Big\|\frac{w}{v}\Big\|_\infty.$$

*Proof.* For operators $A$ and $B$ on $\mathcal{H}$, denote by $A \geqslant B$ that $A - B$ is a non-negative operator. Let $G := \|d\rho_X^{te}/d\rho_X^v\|_\infty$. For all $f \in \mathcal{H}$, we have

$$\langle f, \Sigma f\rangle_\mathcal{H} = \Big\langle f, \int K_x f(x) d\rho_X^{te}(x)\Big\rangle_\mathcal{H} = \int \langle f, K_x\rangle_\mathcal{H} f(x) d\rho_X^{te}(x) = \int f^2(x) d\rho_X^{te}(x)$$

$$= \int f^2(x) \frac{d\rho_X^{te}}{d\rho_X^v}(x) d\rho_X^v(x) \leqslant G\int f^2(x) d\rho_X^v(x) = G\langle f, \Sigma_v f\rangle_\mathcal{H}.$$

Therefore, we have

$$\Sigma \preccurlyeq G\Sigma_v \preccurlyeq G\left(\Sigma_v + \lambda\mathrm{I}\right) \quad\Longrightarrow\quad \Sigma\left(\Sigma_v + \lambda\mathrm{I}\right)^{-1} \preccurlyeq G\mathbb{I},$$

where $\mathbb{I} : \mathcal{H} \mapsto \mathcal{H}$ is the identity operator. This implies $\left\|\Sigma\left(\Sigma_v + \lambda\right)^{-1}\right\| \leqslant G$, which proves the first assertion. The second part comes simply from the fact that

$$\langle f, \Sigma f\rangle_\mathcal{H} = \int f^2(x)\frac{d\rho_X^{te}}{d\rho_X^{tr}}(x)\frac{d\rho_X^{tr}}{d\rho_X^v}(x)d\rho_X^v(x) = \int f^2(x)w(x)\frac{1}{v(x)}d\rho_X^v(x) \leqslant \Big\|\frac{w}{v}\Big\|_\infty \langle f, \Sigma_v f\rangle_\mathcal{H}.$$

$\square$

**Proposition 6** (Cordes Inequality Fujii et al. (1993))**.** *Let $A, B$ two positive semidefinite bounded linear operators on a separable Hilbert space. Then*

$$\|A^s B^s\| \leqslant \|AB\|^s \quad \text{when } 0 \leqslant s \leqslant 1.$$

## D   Experimental Details and Datasets

This section includes some additional details on the experimental setting and a detailed summary of all the datasets presented.

The four datasets consist of data collected from multiple users using wearable sensors, such as accelerometers and gyroscopes. We allocate 1% of the test dataset for weight estimation, reserving the remaining 99% for predictions. Weights are estimated using Relative Unconstrained Least-Squares Importance Fitting (RuLSIF) method, which minimizes the relative density-ratio divergence between distributions (Yamada et al., 2013; Liu et al., 2013).The experiments were conducted in Python on a 2018 MacBook Pro with a 2.3 GHz Intel Core i5 Quad-Core processor, 16GB of RAM, and no GPU.

We give here a brief description of the four datasets.

- The HHAR dataset (Stisen et al., 2015), accessible on DR-NTU (Data), dataset DOI: https://doi.org/10.21979/N9/OWDFXO, version used: 3.0 (May 27, 2022), license: CC BY-NC 4.0, authors of the dataset version: Mohamed Ragab and Emadeldeen Eldele. It comprises 13,062,475 samples with 10 features for human activity recognition using smartphone and smartwatch sensors. It explores the effect of sensor heterogeneity on activity recognition algorithms, aiming to classify physical activities. Data were collected from accelerometer and gyroscope sensors during activities like Biking, Sitting, Standing, Walking, Stair Up, and Stair Down. Nine users participated, using 4 smartwatches (2 LG, 2 Samsung Galaxy) and 8 smartphones (2 each of Samsung Galaxy S3 mini, Samsung Galaxy S3, LG Nexus 4, and Samsung Galaxy S+). Features include accelerometer readings (x, y, z), user ID, device, model, and activity labels. Non-essential columns—index, arrival time, and creation time—were removed. The categorical columns (device and model) were converted to numerical values. User A has 1,218,871 samples designated as training data, from which 15,500 samples were randomly selected, while user H provides the test data.

- The HAR70+ dataset (Ustad et al., 2023), available from the UCI Machine Learning Repository, dataset DOI: https://doi.org/10.24432/C5CW3D, version used: Latest version available via the Norwegian Centre for Research Data (NSD), license: CC BY 4.0 (Creative Commons Attribution 4.0 International), dataset authors: Aleksej Logacjov, Astrid Ustad. It contains 2,259,597 samples with 6 features for activity classification. Data were recorded from 18 older adults (ages 70–95) wearing two Axivity AX3 accelerometers at 50 Hz for 40 minutes in a semi-structured, free-living setting. Five participants used walking aids. Sensors were placed on the right thigh and lower back. Activities such as walking, shuffling, ascending/descending stairs, standing, sitting, and lying were annotated frame-by-frame using chest-mounted camera video. For training and testing, we used data from two files: 518.csv, containing 141,714 samples with 6 features, and 516.csv, containing 138,278 samples with 6 features. From 518.csv, 20,000 samples were randomly selected for training. To introduce covariate shift and study feature selection, the timestamp and thigh accelerations in the x and z directions were excluded.

- The HARChildren dataset (Tørring et al., 2024), available on DR-NTU (Data), dataset DOI: https://doi.org/10.18710/EPCXCC, version used: published on August 30, 2024, license: CC0 1.0, dataset authors: Marte Fossflaten Tørring et al. It contains over 5 million samples with 8 features for activity classification. It includes data from 63 typically developing (TD) children and 16 children with Cerebral Palsy (CP), classified at levels I and II of the Gross Motor Function Classification System (GMFCS). These children wore two accelerometers, on the lower back and thigh. The features initially included a timestamp and accelerometer readings (x, y, z), but the timestamp was removed during preprocessing. The recorded activities include walking, running, shuffling, ascending and descending stairs, standing, sitting, lying, bending, cycling seated, cycling standing, and jumping. For training and testing, two files were used: 004.csv, which contains 354,139 samples, and 010.csv, which contains 238,574 samples. From 010.csv, 15,000 samples were randomly selected for training, while 004.csv was used for testing.

- The WISDM dataset (Kwapisz et al., 2011), available on DR-NTU (Data), dataset DOI: https://doi.org/10.21979/N9/KJWE5B, version used: 1.0 (May 27, 2022), license: CC BY-NC 4.0, dataset authors: Mohamed Ragab and Emadeldeen Eldele. It contains 1,098,209 samples with 5 features from accelerometer and gyroscope data collected at 20 Hz using smartphones and smartwatches. Data were recorded from 36 users performing 18 activities for 3 minutes each. Features include a user ID, activity code (label), and sensor readings (x, y, z). User 12 has 32,641 samples designated as training data, from which 25,000 samples were randomly selected, while user 19 provides the test data.

A common way to specify the grid of possible values of $\lambda$ is to consider a geometric series.

**Remark 1** (Geometric grid). *Let $\lambda_{min}$ and $\lambda_{max}$ the smallest and largest values of the regularization parameter we wish to consider and let*

$$b = \left(\frac{\lambda_{max}}{\lambda_{min}}\right)^{1/(Q-1)}.$$

*We generate a geometric grid of $Q$ values of the regularization if for $q = 1, \ldots, Q$, we let*

$$\lambda_q = b^{q-1}\lambda_{min},$$

*so that $\lambda_1 = \lambda_{min}$ and $\lambda_Q = \lambda_{max}$.*

We used $\lambda_1 = 10^{-4}$, $\lambda_Q = 1$ and $Q = 10$ to generate the geometric grid. We also propose a method for generating a geometric sequence of $\gamma$ values, derived from distinct formulas for even and odd indices.

**Remark 2.** *The following values represent the parameter $\gamma$ for the Gaussian kernel:*

$$\gamma_k = \begin{cases} 10^{-3+\frac{k-1}{2}}, & k \text{ odd}, \\ 5 \cdot 10^{-3+\frac{k-2}{2}}, & k \text{ even}, \end{cases} \quad k = 1, 2, \ldots, 6.$$

We optimize the hyperparameters using hold-out cross-validation, partitioning the dataset into 70% for training and 30% for validation. For each combination of hyperparameters $\lambda$ and $\gamma$, we train the model on $X_{\text{train}}$ and $y_{\text{train}}$ and validate it on $X_{\text{val}}$ and $y_{\text{val}}$.

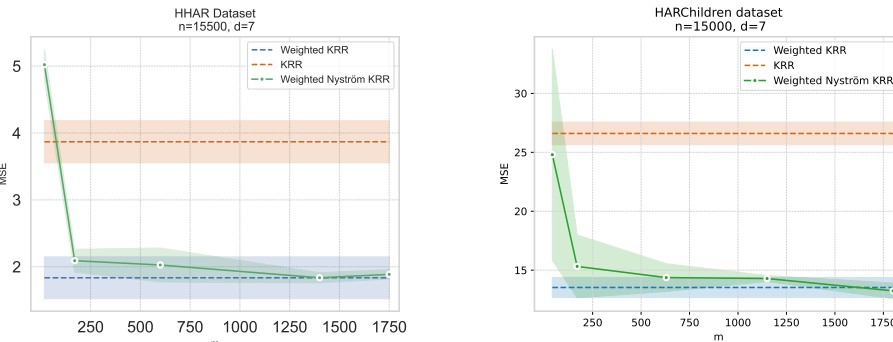

Figure 2: These plots illustrate the results in Table 1 for HHAR and HARChildren datasets.

As regards the simulations, we report here the parameters used to generate the results in Figure 1: $k = 50$ (that we can consider misspecified), $\mu_{tr} = (0.7, 0.7)$, $\sigma_{tr}^2 = \text{diag}(0.7, 0.7)$, $\mu_{te} = (1.8, 1.8)$, $\sigma_{te}^2 = \text{diag}(0.5, 0.5)$, $\epsilon^2 = 0.2$, $c_1 = c_2 = 10$.

Table 2: Performances of the various methods, both in terms of MSE and training/prediction time using uniform sampling.

|  | HAR70+ ($n = 20000$) | | | HARChildren ($n = 15000$) | | | HHAR ($n = 15500$) | | | WISDM ($n = 25000$) | | |
|---|---|---|---|---|---|---|---|---|---|---|---|---|
|  | MSE | $t$ train (s) | $t$ pred (s) | MSE | $t$ train (s) | $t$ pred (s) | MSE | $t$ train (s) | $t$ pred (s) | MSE | $t$ train (s) | $t$ pred (s) |
| KRR | $10 \pm 1$ | $1694 \pm 2$ | $15.0 \pm 0.5$ | $26.6 \pm 0.9$ | $762 \pm 12$ | $10.2 \pm 0.4$ | $3.7 \pm 0.3$ | $876 \pm 6$ | $10.5 \pm 0.9$ | $7.8 \pm 0.1$ | $3280 \pm 48$ | $38 \pm 5$ |
| W-KRR | $5.0 \pm 0.2$ | $1785 \pm 2$ | $15.1 \pm 0.3$ | $13.5 \pm 0.8$ | $809 \pm 26$ | $9.0 \pm 0.1$ | $1.8 \pm 0.3$ | $1034 \pm 93$ | $9.9 \pm 0.1$ | $4.8 \pm 0.2$ | $3364 \pm 30$ | $33 \pm 2$ |
| Ny W-KRR | $5.0 \pm 0.1$ | $67 \pm 0.8$ | $3.8 \pm 0.1$ | $12.9 \pm 0.2$ | $57 \pm 0.2$ | $6.2 \pm 1.0$ | $1.79 \pm 0.01$ | $5.9 \pm 0.1$ | $1.3 \pm 0.2$ | $4.8 \pm 0.1$ | $32 \pm 0.5$ | $7.2 \pm 0.1$ |

Table 2 shows that even when using uniform sampling, the two methods with importance weighting (IW) correction achieve the best and essentially equal performance. However, the number of Nyström points $m$ required by Nyström W-KRR is 5500, 6500, 1750, and 4500 for HAR70+, HARChildren, HHAR, and WISDM respectively, which represents an increase compared to the ALS method.

