# OpenReview forum: "Computational Efficiency under Covariate Shift in Kernel Ridge Regression"
_NeurIPS.cc/2025/Conference — NeurIPS 2025 spotlight_

### Official Review · Reviewer_CabK · 2025-06-17

**Clarity:** 3
**Significance:** 2
**Originality:** 2
**Rating:** 4
**Confidence:** 3

**Summary:**

Under the learning setting where the covariate shift presents, this manuscript investigates the trade-off between computational efficiency and statistical accuracy in kernel ridge regression. Specifically, the authors show the employing random projection techniques, e.g., Nystrom method, in KRR can achieve optimal convergence rates while dramatically reducing computational costs. Studies are extended to unknown IW functions.

**Questions:**

Intuitively, why is Assumption 3 crucial to apply Nystrom method? Do authors think it is necessary? Or it can be relaxed further with different mathematical tools.

**Ethical Concerns:**

["NO or VERY MINOR ethics concerns only"]

**Final Justification:**

The author responses address my concerns. Since my original rating has been positive, I will keep it as it was.

**Limitations:**

Yes. Authors provide a paragraph in introduction to describe the limitations in terms of the theoretical assumptions and their results.

**Quality:**

3

**Strengths And Weaknesses:**

# Strength

1. clear writing, easy to follow.
2. The authors honestly state limitations of their studies, such as the misspecified case they considered, although slightly more general, but conceptually align with well-specified case.
3. Clear discussions about related works, including how assumptions are different and what results are (non-)comparable with existing results.
4. The simulation setting mimics the misspecified setting.

# Weakness:

1. The biggest incentive of consider random projection technique to build KRR estimators is the necessity to compute the inverse of the sample covariance operator, or specifically the matrix in eq. (6). However, KRR is well-known to be a special case of spectral regularized algorithms, including KRR, kernel gradient descent or gradient flow, etc. Therefore, if one generates the estimator via KGD (or others), one can at least avoid the computational time bottleneck of KRR’s matrix inversion.
2. It is interesting to see the strong moment assumption (Assumption 3) over the weight function is crucial for applying Nystron method. While technically explanations are provided in Appendix C, it would be more reader-friendly to provide some intuitive explanations.
3. While having the eigenvalue of covariance operator decay polynomially satisfy Assumption 6, applying Gaussian kernel will not satisfy this assumption. Using Matern kernel with finite order seems more reasonable and align with the assumption.
4. For those who are not familiar with random projection technique in kernel methods, lines 150 - 154 could be a bit overwhelming. That being said, authors still provided enough details in the Appendix. If there is more room allowed, it can be further detailed. Also, missing argmin at the r.h.s. of eq. (18).

---

> ### Author Rebuttal · Authors · 2025-07-29
>
> Thank you for the detailed feedback.
>
> 1. As pointed out, the closed-form KRR estimator requires solving a linear system, which indeed represents the main computational bottleneck.
> First-order iterative schemes such as kernel gradient descent (KGD) or stochastic gradient methods (SGM) avoid matrix inversion and can be computationally advantageous through early stopping.
> Specifically, if the algorithm stops after $T$ iterations, the time complexity is $O(T n^2)$ for gradient descent and conjugate gradient methods [1],[2], and $O(T n)$ for stochastic gradient methods [3].
> These procedures are well understood from a statistical perspective, but they still face significant computational and, more importantly, memory challenges for large datasets, since they require storing and repeatedly multiplying by the empirical kernel matrix, which is at least quadratic in the sample size.
>
> Then, even in this case, a natural way to alleviate these limitations is to replace the full kernel matrix with a low-rank approximation, such as the Nyström method or random features, as widely studied in the literature [4],[5],[6].
> These approximations reduce the storage cost from $O(n^2)$ to $O(n m)$ and the per-iteration cost from $O(n^2)$ to $O(n m)$, where $m \ll n$ is the chosen rank, and can be used not only with KRR but also with iterative schemes like KGD or SGM.
>
> In other words, switching from closed-form KRR to gradient-based solvers does not eliminate the fundamental memory bottleneck associated with storing and manipulating the full kernel matrix; it merely shifts the computational burden.
> Low-rank approximations—Nyström, random features, or random projections—can therefore be beneficial across all spectral regularization methods, regardless of whether they are solved in closed form or via iterative optimization.
>
> ---
>
> 2.  Assumption 3 is needed because, under covariate shift, the importance weights can in principle be unbounded. When applying the Nyström method, we rely on concentration results to control the error of sampling-based approximations of covariance operators. Standard tools such as Bernstein inequalities work well for bounded or sub-exponential random variables, but not for arbitrarily heavy-tailed weights.
>
> Assumption 3 addresses this by imposing a moment condition on the weights: while individual weights are not strictly bounded, the probability of encountering extremely large values is sufficiently small. Intuitively, this prevents rare but extreme weights from dominating the empirical estimates, which could otherwise destabilize the approximation and invalidate the concentration results. From a practical perspective, this reflects the fact that importance weighting works reliably when the shift between training and test distributions is not too extreme (the assumption can also be seen as a condition on the Rényi divergence between test and train distribution, see line 177).
>
> We do not rule out that different mathematical tools—e.g., truncation strategies or robust concentration inequalities etc.—could relax this assumption further, but future work in this direction is needed to understand that. As regards our analysis, Assumption 3 remains crucial and it's not directly relaxable, but it guarantees a good balance between generality and analytical tractability, while being already common in importance-weighted learning literature.
>
> ---
>
> 3. We chose the RBF (Gaussian) kernel because it is widely used in both theory and practice, making it a natural benchmark for our experiments.
> While Assumption 6 requires a polynomial decay of the covariance operator eigenvalues, the RBF kernel actually satisfies an even stronger property—its eigenvalues **decay exponentially rather than polynomially**.  Then Assumption 6 is also satisfied by the RBF kernel.
> Hence, the RBF kernel is fully compatible with our theoretical framework, and our results hold under an even more favorable spectral condition. That said, as the reviewer noted, the Matérn kernel is also a perfectly valid choice within our model.
>
> ---
>
> 4. As also suggested by other reviewers, we will use the extra space to move the details of ALS sampling to the main text.
>
> ---
>
> We tried to answer your question at point 2) .
>
> **References**
> [1] Garvesh Raskutti, Martin J. Wainwright, and Bin Yu. *Early stopping and nonparametric regression: An optimal data-dependent stopping rule.* Journal of Machine Learning Research, 15(1):335–366, 2014.
> [2] Gilles Blanchard and Nicole Krämer. *Optimal learning rates for kernel conjugate gradient regression.* In Advances in Neural Information Processing Systems, pages 226–234, 2010.
> [3] Lorenzo Rosasco and Silvia Villa. *Learning with incremental iterative regularization.* In Advances in Neural Information Processing Systems, pages 1621–1629, 2015.
> [4] Luigi Carratino, Alessandro Rudi, and Lorenzo Rosasco. *Learning with SGD and random features.* Advances in Neural Information Processing Systems 31, 2018.
> [5] Junhong Lin and Lorenzo Rosasco. *Optimal Rates for Learning with Nyström Stochastic Gradient Methods.* arXiv preprint arXiv:1710.07797, 2017.
> [6] Alessandro Rudi, Luigi Carratino, and Lorenzo Rosasco. *FALKON: An optimal large scale kernel method.* Advances in Neural Information Processing Systems 30, 2017.

---

> > ### Comment · Reviewer_CabK · 2025-08-04
> >
> > Thank you for the response.
> >
> > Since my original attitude for this paper has already been positive, I will keep my score and support this manuscript for acceptance.

---

### Official Review · Reviewer_q7kD · 2025-06-29

**Clarity:** 2
**Significance:** 2
**Originality:** 3
**Rating:** 5
**Confidence:** 4

**Summary:**

This paper aims to reduce the computational cost of RKHS-based non-parametric regression under covariate shift between the training and testing distributions. Traditionally, this problem is addressed using kernel ridge regression with importance weighting (IW) correction, which has a computational complexity of $O(n^3)$, making it impractical for large datasets. To address this, the authors propose a Nyström-based approximation method that restricts the search for the minimizer to a low-dimensional random subspace of the RKHS. They prove that the proposed method achieves the optimal convergence rate when the importance weights are known, and they extend the results to the case of unknown weights. Empirical results demonstrate both the efficiency and accuracy of the approach.

**Questions:**

You could address the three points raised under 'Weaknesses' to improve your rating.

**Ethical Concerns:**

["NO or VERY MINOR ethics concerns only"]

**Final Justification:**

I am pleased to raise my score for this interesting paper, which features a solid mathematical foundation and a relevant machine learning application.

**Limitations:**

yes

**Quality:**

3

**Strengths And Weaknesses:**

**Strengths**

* The use of Nyström approximation in the context of non-parametric regression under covariate shift is both relevant and innovative. The paper provides strong statistical guarantees, including non-asymptotic results and effective dimension.
* The related work on kernel ridge regression and its approximations is generally well-situated. However, some notable methods, such as Random Features and SGD, which could also be applicable in this context (especially from a computational standpoint), are omitted.
* The extension of the method to scenarios with unknown IW is interesting.

**Weaknesses**

* Based on Section 6, it remains unclear whether knowledge of the IW  leads to improved convergence rates. Clarifying this would strengthen the theoretical contributions.
* Some important assumptions, such as the requirement that the test distribution is dominated with respect to the training distribution, are not sufficiently highlighted. This assumption rules out support shift and is critical to the theoretical guarantees.
* The misspecified setting (addressed in Section 6) is treated as the harder case, yet there is no discussion of it in Section 5. This separation is confusing, as it leaves the reader uncertain whether the guarantees proven earlier apply only in the well-specified case.

**Writing Suggestions**

* There is redundancy in the presentation of equations, especially in Section 4. Additionally, the definition of the Nyström ALS method should appear in the main. Readers unfamiliar with Nyström methods may find it difficult to understand the approach without it.
* Notation could be made more consistent—for instance, both $\int$ and $\mathbb{E}$ are used for same integration (with $\mu$) in Section 2. Some definitions, such as the norm $\Vert \cdot \Vert_\rho$, are also missing.

In summary, I appreciate the paper's solid mathematical foundation, the clever approach to addressing the problem, and the empirical validation. However, there is room for improvement in terms of clarity and overall soundness.

---

> ### Author Rebuttal · Authors · 2025-07-28
>
> Thank you for the detailed feedback and writing suggestions. We have implemented your advice to improve clarity and readability throughout the paper, uniformizing the notation for expectations and clarifing that $\| . \|_{\rho}$ is simply the norm in $L_2(\rho)$. In particular, as also suggested by other reviewers, we plan to use the additional space to move Appendix B (definition of ALS) into the main text and provide additional commentary to aid readers unfamiliar with Nyström methods.
>
> ---
>
> ### Responses to the Specific Points Raised
>
> 1. **Convergence rates with and without known importance weights (Section 6)**
> Section 6 addresses the more realistic scenario where the importance weights are unknown and must be estimated from data.
> In the well-specified setting (Eq. (21)), consistent with Ma et al. (2023) and Gogolashvili et al. (2023) (both in the full model setting, without Nyström approximation), we show that the convergence rate in Eq. (22) (with estimated weights) matches the rate in Eq. (15) (with exact weights).
>
> As noted in lines 262–264, while the dependence on $n$ is unchanged, the dependence on the infinity norm of $w$ differs and Eq. (22) exhibits a worse dependence. Despite statistical learning theory typically focusing on the dependence on $n$, it is important to note that in practice $\|\|w\|\|_\infty$ can be large under extreme covariate shift, meaning that—even with identical rates—the bound may still deteriorate because of this constant.
> In particular, the interpretation, based on the rate, that IW correction is useless in the well-specified setting can be misleading in applications under severe shift.
>
> In the misspecified setting, lack of knowledge of the true weights is more critical: the resulting estimator is no longer unbiased, and the error generally does not vanish as $n \to \infty$, due to an additional term that remains (lines 265–267). We have clarified these distinctions in the revised version.
>
> ---
>
>  2. **Importance of the absolute continuity assumption**
> We agree that the assumption that the test distribution is absolutely continuous with respect to the training distribution is crucial and deserves to be emphasized.
> This assumption underpins all importance-weighted analyses, as the Radon–Nikodym derivative $w$ must exist to define the weights.
> Intuitively, without overlapping support, the training data cannot provide information about the regression function in regions that only appear in the test distribution.
> We will explicitly highlight the importance of this assumption and its implications in the revised text.
>
> ---
>
>  3. **Well-specified vs. misspecified settings (Sections 5 and 6)**
> The results in Section 5 are proven under Assumption 2, which slightly extends the standard well-specified setting commonly used in the literature.
> As discussed in the Limitations paragraph (lines 88–91), Assumption 2 allows the regression function to lie outside $\mathcal{H}$ but guarantees that its projection onto the hypothesis space $\mathcal{H}$ exists and belongs to its interior rather than its boundary. From the perspective of the technical proofs, this setting is conceptually similar to the standard well-specified case and slightly extends prior work on covariate shift.
>
> The fully misspecified setting (no condition even on the projection) is technically much more challenging and, to our knowledge, is not yet addressed in the covariate shift literature.
> A full treatment of this case (e.g., involving interpolation spaces as in Steinwart & Christmann, 2008) is beyond the scope of our current work but represents an important direction for future research.
>
> Section 6 serves as an additional caution regarding the study of the misspecified setting in the realistic scenario where the true importance weights are unknown. Extending from the well-specified case to the more general misspecified setting, even in the simplified framework under Assumption 2, requires additional care: misspecification not only increases technical complexity, but also introduces practical issues, including potential loss of consistency (lines 265–267). In particular, the predictor $\widehat{f}^v_{\lambda,m}$ learned with approximate weights $v$ is **not a consistent estimator of the target** $f_\mathcal{H}$ anymore, leading to an arbitrarily large error term $\|\|f^v_\mathcal{H} - f_\mathcal{H}\|\|_{\rho_X^{te}}$ when $v$ is a poor approximation of $w$ (see Eq. (20) and discussion at line 266).
>
> We will clarify better in the text the distinction between well-specified (regression function in $\mathcal{H}$), the "simplified" misspecified case under Assumption 2 (the regression function not in $\mathcal{H}$ but it's projection is), and the fully misspecified case (that we do not analyse).
>
> ---
>
> We refer to the answer to reviewer wwtt about the use of other approximation methods such as Random Features or sketching. As regards SGD and early stopping, this can be an interesting alternative to our KRR analysis, as pointed out in the answer to reviewer CabK. We leave these directions to future works for a complete comparison.

---

> > ### Comment · Reviewer_q7kD · 2025-08-06
> > **Thanks.**
> >
> > I thank the authors for considering my suggestions and for their thorough responses to my concerns. I am pleased to raise my score for this interesting paper, which features a solid mathematical foundation and a relevant machine learning application.

---

### Official Review · Reviewer_wwtt · 2025-07-03

**Clarity:** 3
**Significance:** 3
**Originality:** 2
**Rating:** 5
**Confidence:** 3

**Summary:**

This paper tackles the problem of computational efficiency of kernel methods in out-of-distribution (OOD) scenarios. The authors consider importance-weighting (IW) correction to handle distribution mismatch and random projections (the Nystrom method) to improve efficiency, and rigorously analyze the interplay between the two. They extend the previous literature on random-projection methods to the covariate-shift case and propose an efficient algorithm. On realistic datasets, the algorithm is shown to run much faster while achieving similar prediction accuracy to the naive kernel regression.

I could not follow the entire mathematical framework, but I think the general idea is to constrain the solution space by considering only a subset of training data to reduce computational overhead and to apply the IW framework on the restricted space. The authors then prove excess-risk bounds for this algorithm and find that the Nyström method does not really decrease OOD generalization with IW.

**Questions:**

One could alternatively approximate the kernel with random features without subsampling training inputs. How does your method compare to that?

**Ethical Concerns:**

["NO or VERY MINOR ethics concerns only"]

**Final Justification:**

I think this paper's results are solid and potentially impactful. The authors do a good job of explaining the problem clearly, providing a good summary of the previous literature, and stating exactly what their contribution is. With the authors' promised clarifications, I think it will be a highly readable paper, even for non-experts like me.

**Limitations:**

Yes

**Quality:**

3

**Strengths And Weaknesses:**

### Strengths

1. The manuscript is clearly written and well organized; the necessary theoretical concepts are introduced in a clean way.
2. The paper studies two important problems in classical machine learning—efficiency and OOD generalization—simultaneously.
3. The assumptions and limitations are described clearly.

### Weaknesses

1. It was hard for me to follow the analysis after Section 5.
2. The theory assumes ALS to subsample inputs, but its definition is missing. The authors do not discuss how ALS compares to naive subsampling. The choice of subset directly affects IW correction, and it is not clear to me whether ALS is optimal.
3. The paper only considers IW to treat OOD generalization and does not discuss alternative methods.

**Some minor comments/suggestions**

* It may help the reader to increase the font size in the figure.
* In line 300: W-KRR → IW-KRR.
* You could provide a step-by-step description of the algorithm for practitioners.

---

> ### Author Rebuttal · Authors · 2025-07-28
>
> Thank you for the detailed feedback. We already implemented your comments and suggestions to improve clarity and readability of the paper.
>
> Responses to the Specific Points Raised
>
>
> 1.	We refer to the answer to reviewer q7kD to some additional explanation of section 6.
> In case of acceptance, we plan to use the additional page to add explanatory comments and intermediate steps to make the reasoning easier to follow.
>
> 2.    As also noted by other reviewers, we will move the definition of ALS sampling (previously in Appendix B) into the main text and provide additional explanation. Moreover, **we already added an additional table (analogous to Table 1) in the appendix to empirically compare uniform sampling with ALS in practice**. As expected, our results are consistent with previous findings: while uniform sampling can achieve accuracy comparable to ALS, it often requires sampling significantly more Nyström points and, in some cases, even results in higher overall computational cost, effectively nullifying the benefits of its simpler sampling scheme.
>
> We will attempt to provide some explanations regarding 'optimality' below. ALS sampling is well-known among simple independent sampling schemes: Nyström approximation with ALS achieves the same excess risk rate as full KRR using only $m=O(\mathcal{N}(λ))$ Nyström points (see Theorem 1, line 208). Uniform sampling can achieve the same rate but typically requires more points. More sophisticated subset-selection strategies (e.g., adaptive greedy selection, DPPs, pivoted Cholesky) can sometimes use fewer points, but they are computationally more expensive or require additional information (e.g., partial SVD).
> For this reason, ALS sampling is widely regarded as a principled and computationally efficient choice, which aligns with the theoretical scope of our paper (see also answer to reviewer zfAH).
>
> 3) We agree that out-of-distribution (OOD) generalization can be addressed through multiple approaches. In this paper, we focused on importance weighting (IW) because it is one of the most widely used and mathematically well-established methods for covariate shift. IW provides a clear probabilistic framework (via the Radon–Nikodym derivative) and allows for precise statistical guarantees, which aligns with our goal of providing non-asymptotic theoretical results and connecting them with efficient algorithms. We leave as a future direction the exploration of other approaches.
>
> **Random features and alternative methods**
>
> Indeed, a natural extension of our work is to perform a similar analysis under covariate shift using alternative approximation techniques such as sketching or random features. In this first paper, however, we focused on the Nyström method because its data-adaptiveness generally leads to stronger performance: by selecting representative points from the training set, Nyström better captures the geometry of the data. Both theoretical and empirical studies (e.g., Yang et al., 2012, “Nyström Method vs Random Fourier Features: A Theoretical and Empirical Comparison”) have shown that Nyström often achieves superior approximation accuracy for the same computational budget.
>
> That said, extending our theoretical framework to random features and other approximation methods in the covariate-shift setting is an exciting direction for future work.

---

> > ### Comment · Reviewer_wwtt · 2025-08-01
> >
> > I thank the authors for taking the time to address all my concerns. I think this paper makes a solid contribution and is well-written. With the additional experiments and improved clarity, I am willing to increase my score.

---

### Official Review · Reviewer_zfAH · 2025-07-03

**Clarity:** 3
**Significance:** 2
**Originality:** 2
**Rating:** 4
**Confidence:** 3

**Summary:**

This paper tackles the covariate shift problem in nonparametric regression within reproducing kernel Hilbert spaces (RKHSs). While kernel methods offer strong statistical guarantees, their high computational and memory costs limit their scalability. To address this, the authors propose using random projections to define a random subspace of the RKHS, aiming to balance computational efficiency and statistical accuracy under covariate shift. The study demonstrates that this approach can yield substantial computational savings without sacrificing learning performance, even when training and test distributions differ.

**Questions:**

It is well known that the Nyström method can be used to scale up a wide range of computations involving kernel matrices. I am curious how the statistical guarantees presented in this work offer new insights into the practical application of the Nyström method under covariate shift—particularly regarding the choice of sampling scheme, determination of sampling size, and the trade-offs in performance compared to exact computations.  Specifically, while statistical guarantees are valuable, do they shed light on why subsampling often leads to improved performance in practice especially in case of large sample size in terms of predictive accuracy?

I would like to clarify whether all the empirical and theoretical results in this paper are based solely on the approximate leverage score (ALS) sampling scheme. If so, I would encourage the authors to include a comparison with uniform random sampling, particularly in the empirical evaluations. In many cases, random sampling is not only significantly cheaper but can also yield comparable or even better accuracy than ALS (see, for example, Figure 1, left panel in [1]). Including results based on random sampling would provide valuable insights and broaden the practical relevance of the work.

[1] Petros Drineas et al.,  An Experimental Evaluation of a Monte-Carlo Algorithm for Singular Value Decomposition.

**Ethical Concerns:**

["NO or VERY MINOR ethics concerns only"]

**Final Justification:**

I have read through the authors responses, which addressed some of my concerns.

**Quality:**

2

**Strengths And Weaknesses:**

Strengths

The authors derived excess risk bounds for the Nyström predictor defined in Eq. (12), and show that the Nyström approximation does not compromise the state-of-the-art convergence rates, while significantly reducing the computational burden.

Weaknesses

1. I am not sure whether the provided analysis could shed light on practical applications (see questions), and also  in interpreting the empirical results reported in Table~1 where the MSE of Nystrom method in KRR is even lower than without this approximate sampling scheme.

2. The authors employ a more sophisticated sampling strategy using approximate leverage scores (ALS) instead of uniform random sampling. However, in many cases, random sampling can be both more accurate and significantly cheaper than ALS (see related questions for details). In this regard, the analysis in this paper may be less generalizable or practically advantageous than it appears. (see more details in questions).

---

> ### Author Rebuttal · Authors · 2025-07-28
>
> Thank you for these insightful questions.
>
> About the points you raised:
> Our analysis provides new insights in three key directions:
>
> 1.	**New insights in Nyström applications**:
> While the Nyström method is widely used to accelerate kernel computations, most existing analyses assume identical training and test distributions. In contrast, our results explicitly account for covariate shift, including cases with unbounded importance weights, and prove the versatility of the method also in the domain adaptation framework. We show that approximate leverage score (ALS) sampling preserves the optimal convergence rates of full importance-weighted KRR even under covariate shift.
>
> 2.	**Determination of sampling size:**
> Our risk bounds specify how the number of Nyström centers $m$ should scale with the sample size $n$, the regularization parameter, and the complexity of the hypothesis space (via the effective dimension). This gives explicit guidance on how to choose $m$ to achieve the same statistical accuracy as the exact algorithm while reducing time and memory costs by orders of magnitude.
>
> 3.	**Why subsampling can improve accuracy in practice:**
> Theoretically, subsampling does not degrade predictive performance asymptotically. In practice, particularly for very large datasets, the effect the reviewer is mentioning, where accuracy is even improved, can sometimes be attributed to reduced numerical errors and implicit regularization from working in a lower-dimensional subspace. This is consistent with Table 1, where the Nyström approximation achieves accuracy statistically comparable to the exact method despite significantly lower computational cost.
>
> Overall, our work extends Nyström guarantees to a nontrivial covariate shift setting and provides explicit prescriptions for sampling design and computational–statistical trade-offs that were previously unavailable.
>
>
> **Clarification on Table 1 interpretation**
>
> The results in Table 1 are consistent with our theoretical findings:
>
> 1.	Why standard KRR performs worse:
> Standard KRR ignores covariate shift, being trained and evaluated under the assumption that training and test data follow the same distribution. While harmless in well-specified models, this assumption degrades performance in more general misspecified settings. The experiments confirm that methods incorporating importance-weight correction consistently outperform standard KRR.
>
> 2.	Comparison between W-KRR and W-Nyström KRR:
> Our theoretical analysis shows that W-Nyström KRR matches the statistical guarantees of W-KRR while greatly reducing computational cost. Table 1 confirms this empirically: the mean squared error (MSE) of W-Nyström KRR is **statistically indistinguishable** from W-KRR (considering the provided confidence intervals), with small fluctuations attributable to Nyström point selection (via ALS) and cross-validation randomness. Importantly, W-Nyström KRR achieves this accuracy with much lower runtime, which is precisely its intended benefit.
>
> **Uniform sampling vs. ALS**
>
> We agree that uniform sampling is often competitive in practice and is simpler and cheaper to implement.
> However, ALS sampling has stronger theoretical guarantees, leading to tighter error bounds because it adapts to the spectral structure of the kernel matrix. Uniform sampling is agnostic to these properties and typically requires more Nyström points to achieve the same approximation accuracy. As shown by [2], leverage score sampling matches the statistical guarantees of exact kernel ridge regression with fewer centers than uniform sampling, often improving numerical stability and convergence speed (e.g., FALKON-BLESS vs. uniform sampling). In fact, their results show that, although computing leverage scores is more expensive upfront, uniform sampling can ultimately be more costly because it often requires a larger sample size to achieve the same accuracy. This observation is empirically confirmed in [3], where Table 4 shows that, despite its lower sampling cost, uniform sampling frequently needs more Nyström points than leverage score sampling to match accuracy. As a result, even without the overhead of computing leverage scores, uniform sampling can end up being more expensive overall in terms of both runtime and memory.
> Notice also that the mentioned paper by [1] studied a simpler norm-based weighting scheme—not modern leverage score sampling—and in the context of approximate SVD algorithms. A more recent paper of the same author focusing on leverage score sampling and its benefits is instead [4].
>
> Since our paper focuses on theoretical guarantees and validation of those results, we adopted ALS sampling to align with the setting in which these guarantees hold. That said, extending our experiments to include uniform sampling is straightforward, and **we have already added an additional table in the appendix to empirically illustrate this comparison**. As expected, our results are consistent with previous findings: while uniform sampling can achieve accuracy comparable to ALS, it often requires sampling significantly more Nyström points and, in some cases, even results in higher overall computational cost, effectively nullifying the benefits of its simpler sampling scheme.
>
>
> **References**
>
> [1] Drineas, Petros, et al. An Experimental Evaluation of a Monte-Carlo Algorithm for Singular Value Decomposition, LNCS, 2003.
>
> [2] Rudi, Alessandro, et al. On Fast Leverage Score Sampling and Optimal Learning, NeurIPS, 2018.
>
> [3] Della Vecchia, Andrea, et al. Regularized ERM on Random Subspaces, AISTATS, 2021.
>
> [4] Drineas, Petros, et al. Fast Approximation of Matrix Coherence and Statistical Leverage, JMLR, 2012.

---

> > ### Comment · Reviewer_zfAH · 2025-08-06
> >
> > I have read through the authors responses. Overall, I feel that the theories provided in the paper, though highly systematic, do not add too much new insights in such a level as to provide clear guidance on practitioners. (such as how many samples should one use based on your analysis for a specific dataset).
> > I am also wondering whether you could provide numerical comparison of ALS and random-sampling to show on the datasets in this paper, that the statistical guarantees of ALS does translate into practical significance against random sampling (error versus the number of samples used in ALS and random-sampling, and accuracy versus time consumption for each sampling scheme, to justify your statement that efficiency of random-sampling is nullified by the higher sampling ratio it often needs to use.

---

> > > ### Author Response · Authors · 2025-08-07
> > >
> > > We thank the reviewer for their feedback. We would like to emphasize that this paper is primarily theoretical, and the numerical simulations are intended mainly to validate our theoretical findings---specifically, the benefits of importance-weight correction over standard KRR methods (particularly in the misspecified case) and the performance of Nystr\"om approximations compared to full models, which achieve similar accuracy at significantly lower computational cost.
> > >
> > >
> > > The theoretical prescription for the required number of Nyström points $m$ is provided in the statement of Theorem 1 and in Example 1. As for the regularization parameter $\lambda$, it is not directly computable without prior knowledge or estimation of problem-specific parameters such as $\gamma$ or $r$. In practice, practitioners using the Nyström method or other random projection techniques—such as random features or sketching—typically rely on systematic validation procedures to estimate the number of induced points; see, for example, the FALKON algorithm in [Meanti, Giacomo, et al. "Kernel methods through the roof: handling billions of points efficiently." Advances in Neural Information Processing Systems 33 (2020)].
> > >
> > >
> > > Regarding the second part, as mentioned above, the choice of ALS sampling is fundamental for the theoretical analysis because it exploits the potentially favorable spectral structure of the kernel matrix, connecting naturally with the capacity assumption and eventually leading to tighter statistical bounds compared to uniform sampling. When, for some specific data, this capacity assumption is weak, one may expect the advantage of ALS over uniform sampling to be reduced, especially since ALS is computationally more expensive. Systematically benchmarking ALS against uniform sampling in practice is beyond the scope of this work; we refer, among others, to [Rudi, Alessandro, et al. On Fast Leverage Score Sampling and Optimal Learning, NeurIPS, 2018] for a detailed empirical analysis of ALS versus uniform sampling.
> > >
> > >
> > > That said, our experiments are consistent with those previous results. For example, on the WISDM dataset ($n=25000$), ALS selected $1550$ Nyström points, achieving $4.7\pm0.3$ MSE with $9.9\pm0.4$ s training time, whereas uniform sampling required $4500$ points to reach a statistically identical $4.8\pm0.1$ MSE, but took $32\pm0.5$ s. A similar behavior was observed on HARChildren, where ALS again outperformed uniform sampling. On the other hand, for the HHAR dataset, ALS selected $1400$ points, yielding $1.8\pm0.1$ MSE with $6.5\pm0.4$ s training time, while uniform sampling needed only slightly more points ($1750$) to achieve the same error and, given its algorithmic simplicity, resulted in slightly better performance in time ($1.79\pm0.01$ MSE in $5.9\pm0.1$ s). Finally, for HAR70+, the two methods performed comparably.

---

### Decision · Program_Chairs · 2025-09-17

**Decision:**

Accept (spotlight)

**Comment:**

This paper addresses the covariate shift problem in nonparametric regression within reproducing kernel Hilbert spaces (RKHSs).
The authors leverage Nyström method to work with a random subspace of the RKHS.
This allows one to balance between computational efficiency and statistical accuracy under covariate shift.

The main contribution lies in analyzing kernel ridge regression (KRR) under covariate shift, where the authors rigorously characterize the trade-off between computational cost and generalization performance. Notably, they show that using random projections can significantly reduce computational complexity while still achieving optimal convergence rates.

The paper is well-written and clearly motivated. Reviewers appreciated the technical contributions and the relevance of the problem, though they also suggested that the clarity and overall soundness of the presentation could be further improved.
I encourage the authors to take this feedback into account in a future revision.

Overall, this is a solid contribution that provides valuable insights into scalable kernel methods under distribution shift.
I thus recommend acceptance.